# Acceptability and tolerability of repeated intramuscular electroporation of Multi-antigenic HIV (HIVMAG) DNA vaccine among healthy African participants in a phase 1 randomized controlled trial

Juliet Mpendo[1]*, Gaudensia Mutua[2], Annet Nanvubya[1], Omu Anzala[2], Julien Nyombayire[3], Etienne Karita[3], Len Dally[4], Drew Hannaman[5], Matt Price[6], Patricia E. Fast[6], Frances Priddy[6], Huub C. Gelderblom[7], Nancy K. Hills[8]

1 Uganda Virus Research Institute-International AIDS Vaccine Initiative, HIV Vaccine Program, Entebbe, Uganda, 2 Kenya AIDS Vaccine Initiative, University of Nairobi, Nairobi, Kenya, 3 Project San Francisco (PSF), Kigali, Rwanda, 4 EMMES Corporation, Rockville, Maryland, United States of America, 5 Ichor Medical Systems, Inc., San Diego, California, United States of America, 6 International AIDS Vaccine Initiative (IAVI), New York, NY, United States of America, 7 Vaccine and Infectious Disease Division, Fred Hutchinson Cancer Research Center, Seattle, WA, United States of America, 8 University of California at San Francisco, San Francisco, California, United States of America

* jmpendo@iavi.or.ug

## Abstract

### Introduction

Intramuscular electroporation (IM/EP) is a vaccine delivery technique that improves the immunogenicity of DNA vaccines. We evaluated the acceptability and tolerability of electroporation among healthy African study participants.

### Methods

Forty-five participants were administered a DNA vaccine (HIV-MAG) or placebo by electroporation at three visits occurring at four week-intervals. At the end of each visit, participants were asked to rate pain at four times: (1) when the device was placed on the skin and vaccine injected, *before* the electrical stimulation, (2) at the time of electrical stimulation and muscle contraction, and (3) at 10 minutes and (4) 30 minutes after the procedure was completed. For analyses, pain level was dichotomized as either "acceptable" (none/slight/uncomfortable) or "too much" (Intense, severe, and very severe) and examined over time using repeated measures models. Optional brief comments made by participants were summarized anecdotally.

### Results

All 45 participants completed all three vaccination visits; none withdrew from the study due to the electroporation procedure. Most (76%) reported pain levels as acceptable at every time point across all vaccination visits. The majority of "unacceptable" pain was reported at

**Data Availability Statement:** The authors confirm that all the relevant data are within the paper and its Supporting Information files.

**Funding:** This study was made possible by the generous support of the American people through the United States Agency for International Development (USAID) through the International AIDS Vaccine Initiative. Profectus Biosciences, Inc and ICHOR Medical Systems collaborated and reviewed the study design. The salary of Co-author DH was supported by ICHOR Medical Systems. Co-author LD is employed by EMMES Corporation. EMMES employees were involved in the study design, sample size calculation and randomization. The EMMES Corporation did not have any additional role in the study conduct and decision to publish.

**Competing interests:** PEF, MP and FP were employees of IAVI at the time of the study. DH is an employee of Ichor Medical Sciences Inc which owns the rights to the TriGrid Delivery System. The rest of the authors have no competing interests.

the time of electrical stimulation. The majority of the participants (97%) commented that they preferred electroporation to standard injection.

## Conclusion

Repeated intramuscular electroporation for vaccine delivery was found to be acceptable and feasible among healthy African HIV vaccine trial participants. The majority of participants reported an acceptable pain level at all vaccination time points. Further investigation may be warranted into the value of EP to improve immunization outcomes.

ClinicalTrials.gov NCT01496989

## Introduction

Immunogenicity assessment is one of the primary objectives in vaccine development. DNA vaccines, have been found to be weakly immunogenic in humans when delivered intramuscularly [1–3] partly due to low uptake into cells, which in turn results in both insufficient antigen production and poor stimulation of the immune system. Intramuscular electroporation (EP), which increases cell membrane permeability by applying brief electrical impulses to muscle tissue, may overcome this problem, potentially enhancing immunogenicity by 10- to 1000-fold over conventional intramuscular injections. A study in mice using ADVAX DNA-based candidate HIV vaccine delivered by EP showed significantly higher cellular responses [4]. Trials that have been conducted in the United States using HIV-1 DNA vaccine candidate administered via EP with a molecular adjuvant found that electroporation improved immunogenicity and was safe and tolerable [5, 6]. In Africa, studies have also shown that EP has enhanced immune responses. A double-blind, randomized, placebo-controlled phase 1 trial (7) enrolling subjects in Uganda, Rwanda, and Kenya to test an adjuvanted HIV DNA vaccine given intramuscularly by IM/EP found the vaccines to be safe, well tolerated and moderately immunogenic [7]. Another evaluation of vaccine strategy, conducted in Kenya showed that electroporation enhanced immunogenicity to DNA vaccine; overall, however, the benefit of EP was not significant [8].

Electroporation can only be effective and gain widespread usage if perceived by participants as tolerable enough to warrant their returning for a series of vaccinations. A potential deterrent to the acceptability of the electroporation procedure is the pain caused by muscle contraction, which may cause distress or discomfort and discourage participants from completing the vaccine series. Although vaccine studies have reported on the attitudes towards the electroporation procedure itself [6, 9], they have done so across participants in aggregate; none have examined attitudes in the same participants over the course of a series of vaccinations. Most vaccinations will require multiple doses to achieve optimal immunogenicity and efficacy [10, 11]. Therefore, it is important to explore individual participants' attitudes to the level of pain and its acceptability as a series of procedures; strong negative perceptions could impact a patient's willingness to be vaccinated or return for a full vaccination series.

In this study, we utilized data collected in the vaccine trial by Mpendo et al. [7] to conduct a repeated measures analysis of individual perceptions of tolerability over an entire series of EP vaccinations among healthy participants in three African countries. This expands upon the results presented in [7] providing a more detailed characterization of the volunteer experience.

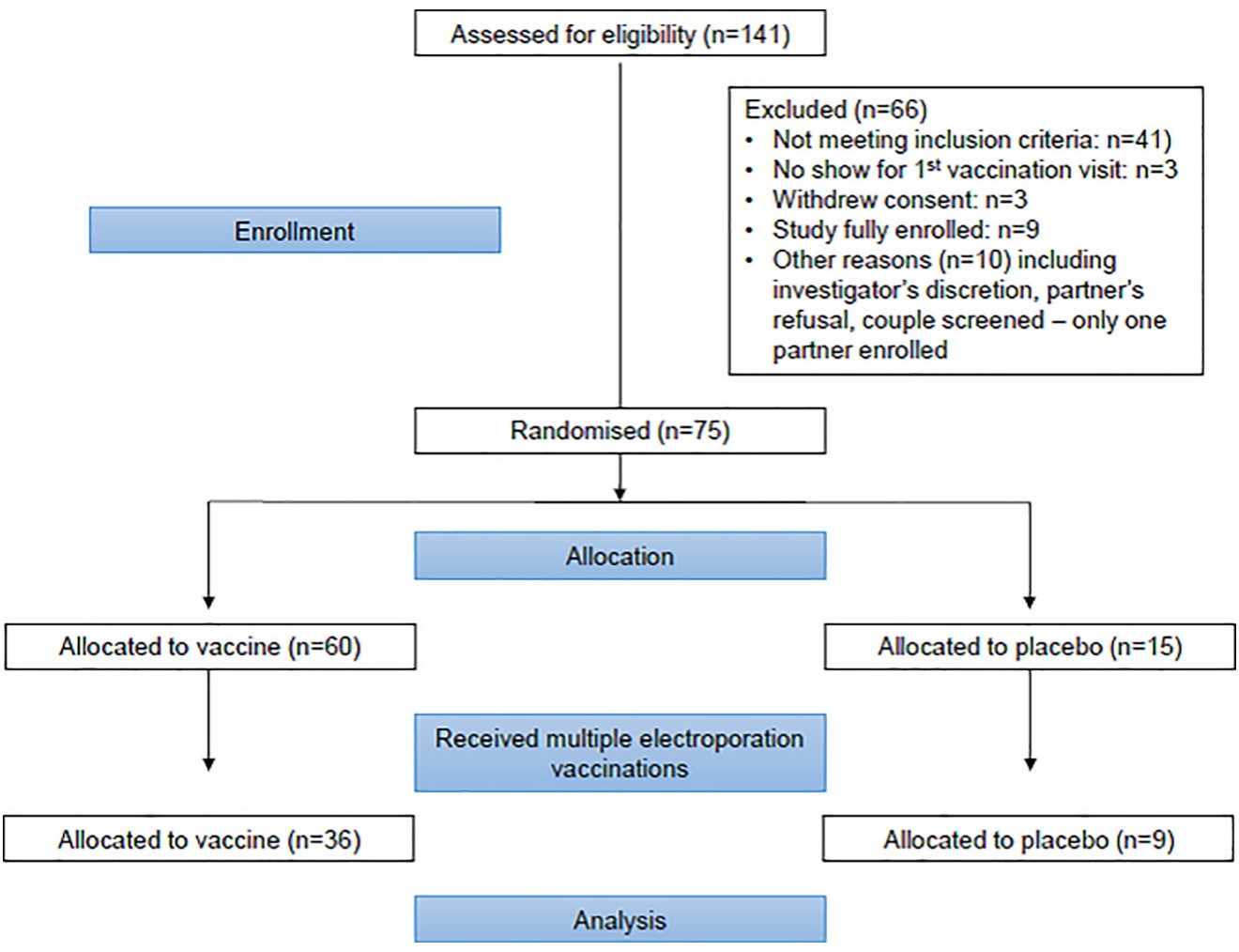

**Fig 1. Consort participant flow diagram.**

## Materials and methods

### Study design and population

The parent study, a multi-center, double-blind, randomized, placebo-controlled phase I trial (ClinicalTrials.gov NCT01496989) designed to evaluate the safety and immunogenicity of heterologous prime-boost regimens, was conducted from December 2011 to March 2013 in urban settings in Uganda, Kenya and Rwanda [7].

As shown in Fig 1, participants, recruited at clinical research centers, included 75 healthy men and women, aged 18–50 years, who were judged to be at low risk for acquiring HIV infection.

Participants were randomly assigned to one of five groups composed of 15 participants each; in three of these groups, participants received the Multi-antigenic HIV (HIV-MAG) DNA vaccine or placebo via electroporation as the prime vaccine at months 0, 1, and 2 and a booster vaccine using a standard injection needle at month 6. There was no deviation to the study protocol. Because we were primarily interested in perception of pain associated with the electroporation in individuals over time and acceptability in terms of compliance with a series of injections, we included in this analysis only participants from the three groups who received multiple electroporation vaccinations (excluding those without longitudinal data), giving us a

sample size of 45. A comparison of the reactions of the 30 excluded subjects to the remaining 45 at the first visit revealed no differences in response between the two groups.

In each of the three groups, 12 received vaccine and 3 received placebo containing sodium chloride. The vaccine recipients received one of three permutations of the (HIV-MAG) DNA vaccine and a molecular adjuvant, GENEVAX® IL-12): in group 1, the DNA vaccine was given without the adjuvant, those in group 2 received vaccine with 100mcg of the adjuvant, while those in group 3 received vaccine with 1000mcg of the adjuvant (Table 1).

Although we hypothesized that the greatest discomfort reported would be due to the electrical stimulation associated with the electroporation procedure and that pain would not vary with the contents of the vaccine, we compared pain levels reported by different groups at each time point to determine whether combining the vaccination and placebo groups was justified.

The electroporation device in Fig 2 has 3 main parts:

**a) The Application Cartridge** is a sterile, single-use component that houses the agent to be delivered (in a standard syringe) and a TriGrid electrode array for the EP procedure. The cartridge is the only subject contact of the system. It encloses 4 electrodes. Prior to administration procedure, the syringe containing the agent is loaded into the application cartridge. Once loaded with the agent, the cartridge is attached to the integrated applicator for administration to the recipient. In order to cater for differences in skin thickness, the cartridge has an adjustable gauge to control injection depth. The syringe and electrodes are not visible to the participants. Once the device is placed on the recipient's tissue site and activated by the operator, the electrodes are deployed into the tissue to the prescribed depth. The EP device first inserts the needle and the agent, then the electrode array which releases the electrical pulses into the tissue at the site of administration causing muscle contractions.

**b) The Integrated Applicator** is a reusable hand held device that houses the single use cartridge. It deploys the electrodes and initiates the administration of the agent at the touch of the activation button.

**c) The pulse stimulator** controls the administration sequences and generates the electroporation pulses. It is connected to the integrated applicator is connected to the pulse stimulator through an incorporated cable.

In this trial, we used the TriGrid Delivery system™, a product of Ichor Medical Systems, for intramuscular injection, for both the (HIV-MAG) DNA vaccine (with or without adjuvant) and the placebo. The small hand-held device has a mechanism that deploys the electrodes and injection needle into the target muscle tissue to administer the vaccine [12].

During the EP procedure, the cartridge loaded with syringe (containing the study products) and needle were inserted into the integrated Applicator and were not visible to the participant. Skin fold thickness of the upper deltoid region, measured using a caliper, was used for setting the Application Cartridge penetration depth to accommodate the varying levels of skin thickness. The range of the needle's depth was 14–40 mm; those whose skin thickness was $\geq 40$ mm were excluded from the trial. Each administration of either (HIV-MAG) DNA vaccine with or without adjuvant or placebo consisted of two injections per vaccination time point, one in the deltoid muscle of each arm. The clinician held the device against the participant's arm for a

**Table 1. Study design.**

| Study group | Dosage | | Intervention group | | |
| --- | --- | --- | --- | --- | --- |
| | Months 0, 1, 2[IM/EP] | Month 6[Injection] | Vaccine (n = 36) | Placebo (n = 9) | Total (n = 45) |
| Group 1 | HIVMAG | Ad35-GRIN/ENV | 12 | 3 | 15 |
| Group 2 | HIVMAG + GENEVAX IL-12 (100μ) | Ad35-GRIN/ENV | 12 | 3 | 15 |
| Group 3 | HIVMAG + GENEVAX IL-12 (100μ) | Ad35-GRIN/ENV | 12 | 3 | 15 |

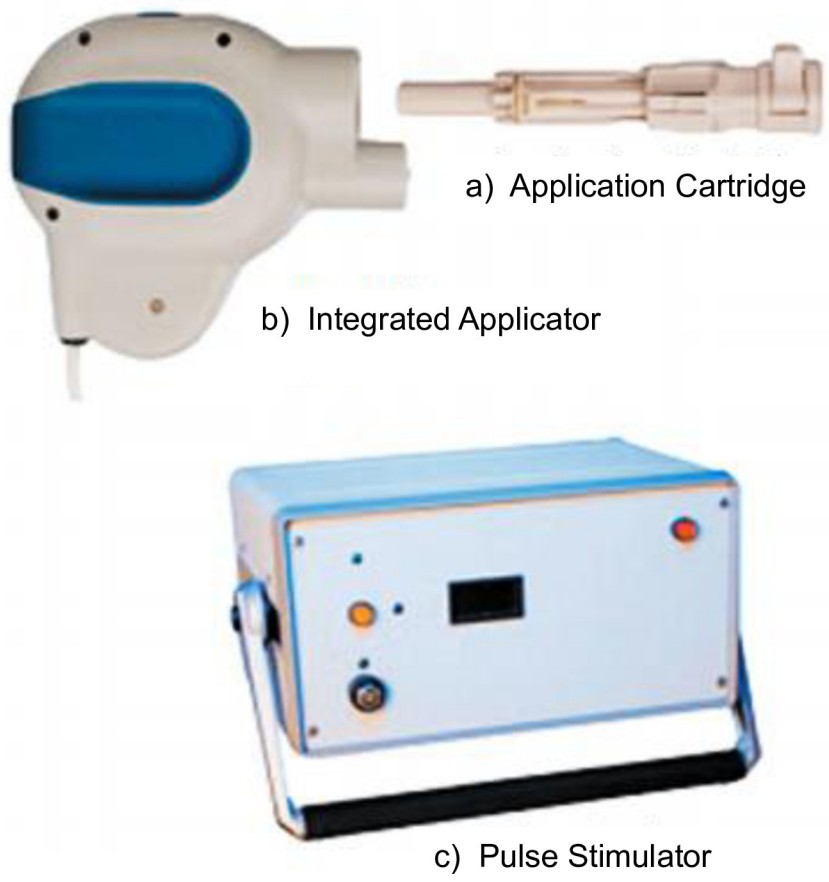

a) Application Cartridge

b) Integrated Applicator

c) Pulse Stimulator

**Fig 2. Electroporation device.**

second or 2, then pressed the activation button on the applicator to initiate the procedure. This was followed by the electrode array which released the electrical pulses into the tissue at the site of administration, causing muscle contractions for 5 to 10 seconds. The procedure was considered complete when the muscle contractions ceased and the appropriate message was received from the pulse stimulator.

During the informed consent process, participants were given details about the electroporation procedure and the involuntary muscle contractions that could potentially cause short term discomfort or pain. In addition, a video showing study physicians undergoing electroporation was shown to potential participants. At screening, height and weight were measured and body mass index (BMI) was calculated.

Ichor provided special training to study physicians on use of the TDS device and addressed correct technique, error codes and management of errors; trained staff were certified on the process. Study physicians blinded as to whether the vaccination contained HIV-MAG DNA or placebo performed all electroporation procedures. Study participants were reimbursed for their time and transport costs.

## Assessments

During the 30–60 minute post-vaccination observation period, a standardized questionnaire was administered to participants by a trained clinical research nurse in either English or the

local language. This face-to-face interview consisted of six questions with the first four questions assessing level of discomfort experienced at different time points: (1) when the device was placed on the skin prior to electrical stimulation, (2) at the time of electrical stimulation and muscle contraction (the duration of the stimulation was very brief, about five seconds, so the response was typically immediately after the stimulation), and (3) at 10 minutes and (4) 30 minutes after the procedure was completed. Participants were asked to rate the pain at the given times as "none", "light", "uncomfortable", "intense", "severe" or "very severe". For purposes of these analyses, we dichotomized pain as either acceptable or too much pain. Because we felt it is reasonable to expect some discomfort with any type of injection, we regarded "uncomfortable" as an acceptable reaction (along with light or none) and compared this to what we considered to be any indication of a level of too much pain that would be unexpected for large scale use (intense, severe or very severe). At each vaccination visit, participants were also asked whether they thought the procedure would be acceptable for vaccine administration in the future if a) it either protected people from getting a serious disease like HIV (for which all volunteers understood that we currently do not have a vaccine), or b) if it contributed to increased scientific knowledge about how to best administer vaccines to prevent infections. Possible responses included "Yes", "No" or "Not sure". We report responses from the final visit only, when the series of vaccinations had been completed. At this point, participants had experienced electroporation and its after-effects three times and were making a decision based on the entire series rather than on one vaccination. Finally, participants were asked if they wished to make a brief general comment about the procedure. All responses were recorded by the study nurses on paper questionnaires, and later entered into an Excel database by data entry officers who were part of the study team. The data manager was responsible for verifying the data entered against the paper source.

## Statistical analysis

Demographic characteristics of the cohort were summarized using means and standard deviations for normally distributed continuous variables and medians and IQRs for continuous variables that were not normally distributed. Categorical variables were summarized using frequencies and percentages. Pain levels reported at each time point across the three vaccine and placebo groups were compared to determine whether combined results for the three groups could be presented. P-values for these tests were calculated using Chi-square and Fisher's exact tests.

Because our analyses included multiple observations per person, we examined the associations between time (across visits) and perception of pain with logistic regression models utilizing generalized estimating equations (GEE) techniques. GEE adjusts the variance of a repeated measures model in order to account for the within-person variability (due to multiple observations per person) in addition to the between-person variability [13]. We ran logistic GEE models, with a logit link and an exchangeable correlation structure with robust standard errors. We first used univariate models to examine whether a participant's perception of the pain associated with electroporation changed across vaccination visits. We analyzed a subject's pain assessment at the time of the injection and at the time of electrical stimulation at visits 1, 2 and 3 (months 0, 1 and 2). This was repeated for the assessments of pain 10 minutes and 30 minutes post-injection, across all visits. We then repeated these analyses, adjusting for potential confounders based on findings from the parent study, including age, gender, BMI, and skin thickness. All analyses were conducted using Stata v14 (College Station, TX). Significance levels were set at an alpha of 0.05.

General comments about the procedure, which were solicited at each visit and either recorded verbatim or summarized by the interviewer, were compiled and summarized

descriptively. These optional comments were brief and were captured on the same questionnaires and are presented as anecdotal evidence supporting quantitative findings.

### Ethical approvals

The study was conducted in compliance with the International Conference on Harmonization for Good Clinical Practice (ICH-GCP). It was approved by all associated Institutional Review Boards including: Uganda Virus Research Institute Research and Ethics Committee, Rwanda National Ethics Committee, Kenyatta National Hospital/University of Nairobi Ethics and Research Committee, the Kenyan national regulatory bodies (National Drug Authority, Director General of Clinical Services Pharmacy and Poisons Board Kenya) and the Recombinant DNA advisory committee, USA. A Safety Review Board consisting of independent clinicians, scientists and statisticians oversaw the progress of the study while monitoring safety and tolerability. Informed consent was obtained before all procedures including electroporation were carried out.

### Partnerships

The study was sponsored by the International AIDS Vaccine Initiative (IAVI), New York, USA. Ichor Medical Systems, Inc., San Diego, California provided the TriGrid™ Delivery system (TDS-IM) and Profectus Biosciences, Inc. manufactured and provided the multi-antigen HIV (HIV-MAG) and GENEVAX® IL-12 vaccine products.

## Results

The majority of the participants were male (60%), with a median (IQR) age of 25 (25, 35), median (IQR) skin thickness fold of 10mm (8, 16) and mean (sd) BMI of 22.6 (5.0).

### Tolerability of pain

We found no statistically significant difference in pain reaction at any time across the four groups (three vaccine and one placebo); therefore, we combined participants from all four groups to compare pain assessments at different times. Each of the 45 subjects attended three vaccine sessions. At each of these sessions, the subjects' reactions were measured at four separate time points relative to IM/EP. Therefore, each subject contributed three "pain ratings" at Time 0 (injection), one at each of the three vaccine visits, for a total of 135 total ratings at Time 0. Similarly, 135 ratings were collected at each of the following: Time 1 (electrical stimulation), Time 10 (10 minutes post IM/EP) and Time 30 (30 minutes post IM/EP).

Of the 45 participants, 34 (76%) reported an acceptable level of pain at every time point across all vaccination visits. The remaining 11 participants (24%) reported 16 instances (12%) of too much pain across 135 visits. Details of reactions are summarized at the level of the individual patient in Table 2.

**Table 2**. No participant reported severe or very severe pain (the two highest pain categories) at any time point during the study. A total of 16 responses of "too much" pain were reported by 11 subjects over the course of the study, with the majority of these reports at the time of electrical stimulation (11/16 = 68.5%). One person reported too much pain associated with electrical stimulation at each of the three visits; two others reported too much pain at two of the three electrical stimulations, and four reported too much pain at the time of electrical stimulation once only (two at Visit 1, one at Visit 2, and one at Visit 3). Three subjects who found the pain at the time of electrical stimulation to be acceptable did report too much pain 10 minutes after the injection, with one of these continuing to report pain 30 minutes after the

**Table 2. Pain assessment relative to timing of electroporation\*.**

| Time point during visit | Injection of agent with needle n (%) | Electrical stimulation n (%) | 10 minutes post IM/EP n (%) | 30 minutes post IM/EP n (%) |
|---|---|---|---|---|
| Vaccine Visit 1 | | | | |
| Acceptable** | 45 (100) | 41 (91.1) | 45 (100) | 45 (100) |
| Unacceptable*** | 0 (0) | 4 (8.9) | 0 (0) | 0 (0) |
| Vaccine Visit 2 | | | | |
| Acceptable** | 45 (100) | 42 (93.3) | 43 (95.6) | 44 (97.8) |
| Unacceptable*** | 0 (0) | 3 (6.7) | 2 (4.4) | 1 (2.2) |
| Vaccine Visit 3 | | | | |
| Acceptable** | 45 (100) | 41 (91.1) | 44 (97.8) | 45 (100) |
| Unacceptable*** | 0 (0) | 4 (8.9) | 1 (2.2) | 0 (0) |

\* Pain assessments at each of the three visits are counted by subject. However, in some cases the same subject reported pain at more than one visit

\*\* Acceptable defined as none, light, or uncomfortable

\*\*\* Unacceptable defined as intense. No instances of severe or very severe pain were reported

injection. An additional subject reported too much pain 30 minutes after injection at the third visit only. Noticeably, although the first time point asked participants to rate the level of pain caused by the initial injection (prior to the electrical stimulation of the IM/EP), no participants at any visit reported too much pain at the time of injection. While it is possible that participants did not experience too much pain from the injection itself, it is also possible that, because electroporation occurred almost immediately after the injection in some cases participants may not have distinguished between the two. However, the fact that injection and electrical stimulation will occur together in the EP process, and that it is the process that is being assessed as acceptable or not, makes distinguishing between the two unnecessary for the purposes of our study. We therefore omitted this first time point (prior to electrical stimulation and muscle contraction) from our longitudinal analyses and utilized the second time point (immediately after electrical stimulation) as a reference point.

One person reported too much pain at the time of EP at each of the three visits, two reported too much pain at two of the three electrical stimulations, and two individuals reported too much pain at the first visit EP only. In every case, reported pain had decreased by 10 minutes post-injection. The remaining five reports were associated with post-electroporation pain in participants who had reported that the pain at the time of the electroporation itself was acceptable.

In univariate logistic regression models utilizing GEE techniques, we examined each subject's reactions over time to the same time point; e.g., a subject's response to electrical stimulation at Visit 1, Visit 2, and Visit 3. Likewise, we examined individual reactions across time at 10 minutes post IM/EP, and separately at 30 minutes post IM/EP. At the time of electrical stimulation, no significant difference was found in subject responses across visits. While the small size of our study does not rule out the possibility of having insufficient power to detect small differences, our results at this time point (where most "intense" reactions were reported) are supported by the fact the 72% of the cohort gave an acceptable rating at the time of electrical stimulation across all three visits. Therefore, a majority of subjects found the electrical stimulation itself acceptable, and this opinion did not change over time.

In evaluating reactions 10 minutes post IM/EP and 30 minutes post IM/EP across visits, we were unable to use the first visit as a reference as no subjects at that visit rated the pain at either 10 or 30 minutes post-injection to be unacceptable. This reduced our power to detect a difference between responses at Visit 3 using Visit 2 as a reference. While subjects appeared less

**Table 3. Logistical regression models of association between perception of acceptable pain and visit number for three vaccination time points.**

| Time point at each of the three visits | Unadjusted logistical regression models | | | Adjusted logistical regression models* | | |
|---|---|---|---|---|---|---|
| | Odds ratio | 95% CI | p-value | Odds ratio | 95% CI | p-value |
| Electrical stimulation** | | | | | | |
| Vaccine visit 1 | Ref | | | Ref | | |
| Vaccine visit 2 | 0.73 | (0.18, 2.92) | 0.66 | 0.71 | (0.13, 0.82) | 0.69 |
| Vaccine visit 3 | 1.00 | (0.34, 2.96) | 1.00 | 1.03 | (0.27, 3.86) | 0.97 |
| 10 minutes post | | | | | | |
| Vaccine visit 1 | n/a*** | | | n/a*** | | |
| Vaccine visit 2 | Ref | | | Ref | | |
| Vaccine visit 3 | 0.48 | (0.04, 5.97) | 0.58 | 0.47 | (0.03, 6.52) | 0.57 |
| 30 minutes post | | | | | | |
| Vaccine visit 1 | n/a*** | | | n/a*** | | |
| Vaccine visit 2 | Ref | | | Ref | | |
| Vaccine visit 3 | 1.00 | (0.06, 17.6) | 1.00 | 1.01 | (0.03, 31.1) | 1.00 |

* Adjusted for age, gender, skin thickness and BMI

** As noted, this may represent a combined reaction to the electrical stimulation and the injection that immediately preceded it

*** There were no reports of "unacceptable" pain at visit 1 for these time points

likely at Visit 3 compared to Visit 2 to report an unacceptable level of pain at 10 minutes post IM/EP, this difference was not significant and confidence intervals were wide. At 30 minutes post IM/EP, no evidence was seen of a difference between reports of unacceptable pain at Visit 3 compared to Visit 2, but again, confidence intervals were quite wide for these comparisons. Adjusted models yielded results that were very similar to those from the unadjusted models (Table 3).

There was little variation in an individual's response between the actual time of vaccination and 30 minutes post vaccination. In all cases, those who reported too much pain at the time of electrical stimulation reported that this pain had lessened to an acceptable level by 10 minutes post-injection. Only in four cases did participants report an increase from acceptable pain at the time of vaccination to too much pain at time points post-injection. Two of these reported intense pain at 10 minutes post-injection that had resolved by 30 minutes post injection, one reported intense pain at both 10 minutes and 30 minutes post-injection, and one reported intense pain at 30-minutes post-injection only.

## General comments about electroporation

At each visit, before leaving, participants were asked if they had any comments about the procedure. Thirty-nine (87%) participants responded at least once during three vaccination visits; several themes were identified from these comments.

## Preference of electroporation over conventional injection

Although participants were not specifically asked to compare electroporation to standard injections, nearly half (18/39) commented that they preferred the electroporation procedure to the injection for a variety of reasons. Several participants cited what they felt was a psychological advantage:

> "*It is not scaring because its needles are not visible;" ". . .as you do not see the needle, psychologically you feel more comfortable;" ". . . the volunteer does not see the needle therefore there*

*is no fear of feeling pain.*" Physical advantages were also perceived: "*EP is faster since it lasted a few seconds and less painful than injection;*"; "*It is better than being vaccinated with a needle. The needle is so painful while with electroporation [you] only feel the arm shake;*" "*The electroporation is smoother and faster than the normal injection.*"

### Perception of electroporation as scary

On the other hand, a little more than a quarter (11/39) of the participants reported that the muscle contraction was "scary." Typical responses included: "*The muscle contraction was stressing and scary at the first time;*" "*The muscle jerking is a little scary;*" "*. . .If there was a way to reduce on the electrical stimulation and hence the contraction, that would be better*" and"*Many people may fear the stimulation part of the procedure.*"

### Pain during electroporation

Only 15% (6/39) of participants commented that they found electroporation to be painful. For example, one participant "*experienced a painful injection on the right arm.*" Another found the vaccination painless but felt "*some pain later.*" However, 17/39 (44%) participants reported the pain to be minimal: "*. . .the pain was not as much as [I] anticipated;*" "*it is fast and the pain minimal;*" "*Electroporation is less painful than the usual injection;*" "*It is better compared to using a needle to vaccinate because EP does not have much pain.*"

### Electroporation perceived as safe

A number of participants (8/39) thought that electroporation was safe in terms of maintaining sterility and reducing the potential risk of infections: "*It is fast and it will prevent in cross infection;*" "*It is safe it cannot be contaminated easily;*" "*one cannot make a mistake when administering it compared to being injected with a needle.*"

### Acceptability of electroporation

Participants were asked at each visit if they felt that the electroporation procedure would be acceptable if it provided protection from serious disease. After the first vaccination, 44/45 of the participants responded positively, with one uncertain. After the second vaccination, 43/45 responded positively, with one each responding "no" and "uncertain." After the third vaccination, 43/45 responded positively, with the remaining two responding "no." Responses were similar when participants were asked if the electroporation procedure would be acceptable if it enhanced scientific knowledge about how best to administer/treat a disease. After the first vaccine, 44/45 responded positively, with one uncertain. After the last vaccine, 42/45 responded positively, with two responding "no" to both questions and one "uncertain." The two subjects who responded "no", as well as the subject who was "uncertain" about the second question only, had both reversed previously affirmative answers for both questions, although neither had reported "too much" pain at any time point or visit.

### Discussion

In this study, we provide a more detailed summary of the volunteer experience following repeated intramuscular electroporation in an HIV vaccine clinical trial [7]. We found that electroporation was acceptable and study compliance and retention feasible among healthy adult African participants. While we cannot ascribe the complete follow-up we observed wholly to the acceptability of the electroporation process, we feel that perfect attendance does support

the idea that the procedure itself was tolerable enough that subjects were not discouraged from returning to undergo it a second and third time.

Only 11 out of 45 (24%) participants reported too much pain at any time point across the three visits, and these were primarily associated with the electrical stimulation/muscle contraction itself. Additionally, no volunteers reported "severe" or "very severe" (the two most extreme categories of pain) at any point during the study. Seven subjects reported intense pain at the time of electrical stimulation (one patient reported intense pain at the time of electrical stimulation at each of the three visits; two more reported this at two visits each); in all 11 instances, however, the pain had lessened to an acceptable level by 10-minutes post injection.

Similar findings were reported in other studies [5, 6] where participants also reported that pain was reduced within 25 and 30 minutes after vaccination. In another study using a dermal EP system, pain had significantly decreased within 5 minutes [14]. Most of the participants stated that they did not mind electroporation. The primary concern reported was that the procedure seemed to be scary, although this effect was largely confined to the first visit. This was likely an emotional response to a new and "unknown" procedure, since the physical response, as rated quantitatively, was more often reported to be "intense" at some time point during visits 2 and 3 than at visit 1.

Our results show that repeated vaccinations using electroporation were almost universally acceptable, with the majority of "unacceptable" pain reports associated with the electrical stimulation which occurred as a part of the procedure. No major differences in pain were reported within individuals across visits, and all participants returned as required and completed the entire vaccination series that consisted of a total of four visits: three visits at which the electroporation procedure was used to administer the vaccine, and one visit (the last) at which needle injection was used rather than electroporation. This finding is similar to a study in the United States [15] that showed no difference in level of pain reported after electroporation procedures that were 14 weeks apart, and with majority of the participants returning for the second electroporation.

The consistently low level of pain reported by participants in our study was similar to that reported in other EP vaccine tolerability studies that have been conducted in the western world [5, 6, 9, 14–16]. However, in one trial in Africa [8], much as EP was well tolerated, one participant who received the vaccine using the EP method reported severe pain after the procedure.

In this study, almost half the participants reported that they preferred electroporation system to the standard injection. Similar results were reported in another study where participants preferred EP compared to the traditional injection though EP was given by a dermal electroporation system [14]. Some participants reported that initially the EP seemed scary, with similar comments reported in another study [14].

The majority of participants responded that they would accept a vaccine given by electroporation if it protected people from getting a serious disease such as HIV for which there is currently no licensed vaccine and/or if it contributed to increased scientific knowledge about how best to administer vaccines to prevent or treat infections. Similar responses were reported by participants in other studies that used different devices; CELLECTRA, Inovio Pharmaceuticals and Easy Vax$^{TM}$ clinical EE device [5, 6, 14]. However, in one of them, when participants were asked whether they would undergo electroporation if the technique was used to increase the effectiveness of an already existing vaccine, the acceptability was lower [5].

In our trial, some vaccine administrations were done by traditional injection, without the EP device. One of the limitations of our study was that we did not assess pain after administering the booster using the standard injection; therefore, any comparisons between the two

techniques for administering the vaccine (EP vs traditional injection) are based on historical experience as reported by participants and not on data recorded at the time of injection. In addition, assessment of pain was based on participants' self-report. Our sample size was modest, limiting our ability to conduct multivariable analysis. Although we had some limited comments on the procedure, these comments were optional and not offered by all participants. Extensive qualitative assessment was not possible since neither focus groups nor interviews were conducted as a part of the parent study. In addition, we cannot be certain as to the generalizability of the study, since study participants participated in a controlled research environment with ample opportunities for questions, comments, and explanations of the process. In a real-world situation, it is unlikely that this degree of support would be available.

However, our ability to examine relative changes over time was a strength of our study; we were able to assess levels of pain across three different vaccination visits and therefore provide more insight into the acceptability and feasibility of compliance with repeated vaccinations. In addition, participants were questioned immediately after the designated time points, therefore eliminating any recall bias. Comments were solicited to supplement the scaled levels of pain reported. Lastly, there was no loss to follow up; 100% of participants returned for all the vaccinations.

In conclusion, we found in our study that repeated administration of DNA vaccines by electroporation was acceptable and well tolerated, supporting the idea that this procedure could be adopted for future vaccine administrations. Novel, improved methods of vaccine delivery are needed for to assure durable immune responses with maximum efficacy, and will likely require a series of vaccinations;. Electroporation, which has been shown to be tolerable, acceptable, and even preferable to conventional injections, provides a promising technology for enhancement of immune responses of DNA vaccines.

## Supporting information

**S1 Dataset.**
(XLSX)

**S1 File.**
(PDF)

**S2 File.**
(DOC)

## Acknowledgments

We would like to acknowledge the contributions of the study volunteers as well as the following individuals on the B004 study team: Jim Ackland, Devika Zachariah, Eddy Sayeed, Apolo Balyegisawa, Kundai Chinyenze, Michele Fong Lim, Paramesh Chetty, Carl Verlinde, Dani Vooijs, the study staff at Kenya AIDS Vaccine Initiative, Uganda Virus Research Institute-IAVI, MRC Uganda, Projet San Francisco Rwanda, and Lorna Clark, Francesco Lala, Laura Sharpe, Jana Carga, Emmanuel Cormier and members of the IAVI Human Immunology Laboratory, Imperial College, London, UK. Also thank you to Burc Barin from EMMES, Inc. for statistical analysis, Jean-Louis Excler (US Military HIV Research Program, Bethesda MD, USA), Christopher Parks (Development and Design Laboratory, IAVI, NY, USA) and Pervin Anklesaria (Gates Foundation, WA, USA) for their contributions to the design of the study. We wish to acknowledge the support from the University of California, San Francisco's International Traineeships in AIDS Prevention Studies (ITAPS), U.S. NIMH, R25MH064712.

## Author Contributions

**Conceptualization:** Patricia E. Fast, Frances Priddy.

**Formal analysis:** Len Dally, Matt Price, Nancy K. Hills.

**Funding acquisition:** Frances Priddy.

**Investigation:** Juliet Mpendo, Gaudensia Mutua, Omu Anzala, Etienne Karita, Patricia E. Fast.

**Methodology:** Annet Nanvubya, Julien Nyombayire, Drew Hannaman.

**Project administration:** Juliet Mpendo, Omu Anzala, Etienne Karita.

**Supervision:** Juliet Mpendo, Annet Nanvubya, Omu Anzala, Etienne Karita.

**Writing – original draft:** Juliet Mpendo.

**Writing – review & editing:** Juliet Mpendo, Gaudensia Mutua, Annet Nanvubya, Omu Anzala, Julien Nyombayire, Etienne Karita, Matt Price, Patricia E. Fast, Frances Priddy, Huub C. Gelderblom.

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
