## [Decision Letter · Decision Letter 0]

2 Oct 2019

PONE-D-19-22032

Acceptability and tolerability of repeated intramuscular electroporation of Multi-antigenic HIV (HIVMAG) DNA vaccine among healthy African participants in a phase 1 randomized controlled trial.

PLOS ONE

Dear Dr MPENDO,

Thank you for submitting your manuscript to PLOS ONE. After careful consideration, we feel that it has merit but does not fully meet PLOS ONE’s publication criteria as it currently stands. Therefore, we invite you to submit a revised version of the manuscript that addresses the points raised during the review process. In particular, please pay special attention to Reviewer #2's comments regarding the statistical analysis.

We would appreciate receiving your revised manuscript by Nov 16 2019 11:59PM. To enhance the reproducibility of your results, we recommend that if applicable you deposit your laboratory protocols in protocols.io, where a protocol can be assigned its own identifier (DOI) such that it can be cited independently in the future. For instructions see: http://journals.plos.org/plosone/s/submission-guidelines#loc-laboratory-protocols

We look forward to receiving your revised manuscript.

Kind regards,

David Joseph Diemert, M.D.

Academic Editor

PLOS ONE

**Journal Requirements:**

a) Did participants provide their written or verbal informed consent to participate in this study?

**Comments to the Author**

1. Is the manuscript technically sound, and do the data support the conclusions?

Reviewer #1: Yes

Reviewer #2: Partly

Reviewer #3: Yes

2. Has the statistical analysis been performed appropriately and rigorously? 

Reviewer #1: Yes

Reviewer #2: No

Reviewer #3: Yes

3. Have the authors made all data underlying the findings in their manuscript fully available?

Reviewer #1: Yes

Reviewer #2: Yes

Reviewer #3: Yes

4. Is the manuscript presented in an intelligible fashion and written in standard English?

Reviewer #1: Yes

Reviewer #2: Yes

Reviewer #3: Yes

5. Review Comments to the Author

Reviewer #1: Mpendo, et al, have presented a subset of data from a previously performed clinical trial assessing the acceptability of intramuscular electroporation as a delivery method for DNA vaccination. Researchers continue to explore methods to enhance the immunogenicity of DNA vaccines and electroporation is increasingly touted as a potentially viable method. While it may be determined that electroporation sufficiently increases immunologic responses, it will not be an effective vaccination method if patients are unwilling to undergo the procedure. Results such as these will be viewed as important in determining whether or not to continue to explore this method of delivery.

The authors have done a fine job presenting their data. The manuscript is well written and clear. I have only minor questions or comments:

The parent study was 2011-2013. That is a significant delay in publishing these results. Can the authors comment on why this is now seen as important enough to publish?

There were 75 people in the original study. They only included 45 in this study (the three groups who received multiple EP injections). They note in the Discussion that these data were not collected on patients receiving the vaccine by standard administration. Was this same information gathered on any others in the study and just not included in this analysis? If so, would the results change if they were included?

Please clarify: When the device was held against the participant’s arm for a second or 2, were needles inserted into the arm at that point or were they triggered (“fired”) by pressing the activation button? I assume from the patients comments that the needles were not initially visible. Was this one needle? Were there 4 electrode needles? 5 needles in total? The skin thickness was measured, but these were IM injections. How deep did the needles go? Some positive patient comments included statements about reducing potential for error and contamination? Can I assume that the EP attachment is single use and disposable? Would it be possible to present a picture or diagram of the device for clarification?

Reviewer #2: Summary: The analyses for this study are reasonably well done, but there is basically zero statistical power in these analyses. Hence, I question the utility of some of these analyses as I see the conclusions for some as misleading and potentially harmful.

1. (lines 144-145) I understand the allure of dichotomizing outcomes, but research suggests the loss of power after dichotomizing is large (e.g., https://doi.org/10.1002/pst.331). This seems like a really good opportunity to analyze the quantitative values. That will be a more powerful analysis and will provide a sense of how much the mean changes over time.

2. (lines 167-174) I found this write up a little confusing. It sounds like logistic regression models are run at each time point and then a marginal model (using GEE) is also implemented. Are all time points included in the GEE-based model?

3. (line 171) Please provide a methodological citation for GEE.

4. Please indicate the level of significance and the software package used in these analyses.

5. (lines 306-308) I think you need analyses specifically targeted at this. I may have missed something, but I don't think you have specifically tested this question in the manuscript. The modeling in this study has shown that perceptions of pain don't change over time, not that the levels of pain were acceptable and feasible. The proportion of participants reporting "too much pain" is presented without any sort of testing. I believe you will need to present some non-inferiority testing against a margin. Maybe in a non-trial setting 24% is not good and hinders a vaccination campaign.

6. (lines 308-310) I don't think the fact that everyone completed the series is robust evidence. There is the potential for compliance and reporting bias in trials, which makes this not the best justification.

7. (lines 313-315) Again, and apologies if I missed this, but I think you need to perform analyses specifically targeted at this question.

8. (lines 322-325) While true that there were no major differences, the power to find a difference is basically nothing here. In Table 3, the 10 minutes post for visit 3 confidence intervals stretch from ~ 0.05 to ~6.00. At best, these models are essentially worthless and tell us nothing. At worst, conclusions are based off these results which basically have minimal chance of finding a difference. That's what is happening here. These conclusions are very misleading. Maybe if this study was a proof of concept study, then I would be more lenient, but you've stated that there are already other studies showing the same thing.

9. (lines 338-345) The survey or qualitative information you provide here may be the most interesting and novel about this manuscript. You might consider restructuring this paper and making it a qualitative manuscript.

Reviewer #3: The manuscript “Acceptability and tolerability of repeated intramuscular electroporation of Multiantigenic HIV (HIVMAG) DNA vaccine among healthy African participants in a phase 1

randomized controlled trial” by Mpendo et al describes the results from an ancillary study assessing the individual perceptions of tolerability over series of intramuscular (IM) electroporation (EP) vaccinations among healthy participants in three African countries, Uganda, Kenya and Rwanda. The parent study, also by Mpendo et al, was a multi-center, double-blind, randomized, placebo-controlled phase I trial, conducted between 2011-2013, assessing the safety and immunogenicity of heterologous prime-boost regimens using a multi-antigenic HIV (HIV-MAG) DNA vaccine or placebo as the prime vaccine at months 0, 1, and 2 with or without adjuvant (GENEVAX IL-12) and a booster vaccine using Ad35-GRIN/ENV or placebo. This ancillary study included 45 of the 75 subjects enrolled in the parent study (subjects who received EP immediately after the injection with the prime vaccine or placebo). The authors provide data and analyses for the research questions that are clearly summarized and represent important information for the vaccine field. I have no major comments or concerns regarding publication of this manuscript. I have minor recommendations (listed in order of appearance):

Affiliations:

- Please list affiliations in numeric order

Abstract:

- Line 25: Please include “..intramuscular (I.M.) electroporation”

- Line 41: Should read “outcomes”?

Partnerships:

- Line 195: Is "system" duplicated

Results:

- Line 205: "...combined participants from all four groups..". Please revise.

- Line 208: 16 instances correspond to 12%. Please revise.

- Lines 230-232: From the text description you seem to only have 14 reports of "too much" pain vs the 16 reported in the table and text above.

- Lines 230-231: Please clarify if there were 5 participants who experienced "too much" pain after post-electroporation or there were fewer participants with repetitive events.

- Lines 247-249: Your statement here is not aligned with you statement in lines 229-230 (on the first you say "in the majority of cases" and on the second you say "In every case"). Please include the remaining reported events of "too much pain" in lines 230-232 to ensure consistency.

Acceptability of electroporation

- You have only analyzed responses from the final visit, after the series of vaccinations had been completed. In a real life scenario it would be important to consider the perceptions after the first vaccination/EP since these may have an impact in retention. Please consider including this analysis and comparing the responses between the first and last visit. Also note that your study participants were under a very controlled research environment with possible extensive counselling which will differ in a real life scenario.

Discussion:

- Please comment on whether the grading of pain at the first visit may have been influenced by the fact that it was a complete new experience and the fear related to the procedure could have also played an important role.

- Lines 324-325: "... three electroporation procedures and one needle injection". It is one injection per electroporation. Please revise.

6. PLOS authors have the option to publish the peer review history of their article (what does this mean?). If published, this will include your full peer review and any attached files.

Reviewer #1: No

Reviewer #2: No

Reviewer #3: No

---

## [Author Response · Author response to Decision Letter 0]

20 Nov 2019

11th Nov 2019

David Joseph Diemert

Academic Editor

PLOS ONE

Dear Dr. Diemert,

We are delighted that PLOS ONE considered our manuscript for review. We have given careful consideration to all the comments from the reviewers and editor, and have addressed them all. 

The following is a point by point explanation of how we have addressed the concerns and revised our manuscript. The line number(s) on the revised manuscript with Track changes have been specified to show the text representing each response.

Comments from Reviewer 1 

Mpendo, et al, have presented a subset of data from a previously performed clinical trial assessing the acceptability of intramuscular electroporation as a delivery method for DNA vaccination. Researchers continue to explore methods to enhance the immunogenicity of DNA vaccines and electroporation is increasingly touted as a potentially viable method. While it may be determined that electroporation sufficiently increases immunologic responses, it will not be an effective vaccination method if patients are unwilling to undergo the procedure. Results such as these will be viewed as important in determining whether or not to continue to explore this method of delivery.

The authors have done a fine job presenting their data. The manuscript is well written and clear. I have only minor questions or comments:

1. The parent study was 2011-2013. That is a significant delay in publishing these results. Can the authors comment on why this is now seen as important enough to publish? 

Response: Novel methods of HIV vaccine administration that will provide good immunogenicity still need to be explored. Although the parent study was conducted from 2011-2013, the website ClinicalTrials.gov currently reports a number of ongoing and recruiting trials utilizing electroporation DNA vaccine delivery. We therefore believe that our data can still provide valuable information, especially for recruiting purposes, in helping to assure potential trial subjects of the acceptability of the process. Because we had the opportunity to assess the reactions to the electroporation process in individuals over the course of time, this can be offered as support that the process was not only acceptable to subjects on a one-time basis, but acceptable enough to guarantee their return for multiple procedures. 

2. There were 75 people in the original study. They only included 45 in this study (the three groups who received multiple EP injections). They note in the Discussion that these data were not collected on patients receiving the vaccine by standard administration. Was this same information gathered on any others in the study and just not included in this analysis? If so, would the results change if they were included? 

Response: Data on pain/acceptability were collected on all 75 participants upon receipt of EP. The parent study assessed pain/acceptability in aggregate at each time point. They found no statistically significant difference in proportions of volunteers who rated the procedure uncomfortable or intense at any time point among the five vaccine and one placebo groups.1 Our analysis is unique in that we looked at responses to the EP procedure within individuals across the series of vaccines. This allowed us to ascertain whether reactions to repeated injections over time were acceptable within each individual. We therefore excluded from our analyses the 30 participants who only received EP at one time point, as they could not add any information to our longitudinal analysis. A comparison of these 30 to the remaining 45 at the first visit revealed no differences in response between the two groups (EP once only vs. EP multiple times). We have clarified this in the text and added information (lines 89-94):

…we included in this analysis only participants from the three groups who received multiple electroporation vaccinations (excluding those without longitudinal data), giving us a sample size of 45. A comparison of the reactions of the 30 excluded subjects to the remaining 45 at the first visit revealed no differences in response between the two groups.

1Mpendo J, Mutua G, Nyombayire J, Ingabire R, Nanvubya A, Anzala O, et al. A Phase I Double Blind, Placebo-Controlled, Randomized Study of the Safety and Immunogenicity of Electroporated HIV DNA with or without Interleukin 12 in Prime-Boost Combinations with an Ad35 HIV Vaccine in Healthy HIV-Seronegative African Adults. PloS one. 2015;10(8):e0134287.

3. Please clarify: When the device was held against the participant’s arm for a second or 2, were needles inserted into the arm at that point or were they triggered (“fired”) by pressing the activation button? 

Response: The insertion of the needle is triggered by pressing the activation button. We thank the reviewer for pointing out the need for further clarification on the EP process. We have briefly described the TriGrid delivery system, which is comprised of three components, to more clearly illustrate the procedure. The text below has been inserted into the manuscript (lines 109-125)

a) The Application Cartridge is a sterile, single-use component that houses the agent to be delivered (in a standard syringe) and a TriGrid electrode array for the EP procedure. The cartridge is the only subject contact of the system. It encloses 4 electrodes. Prior to administration procedure, the syringe containing the agent is loaded into the application cartridge. Once loaded with the agent, the cartridge is attached to the integrated applicator for administration to the recipient. In order to cater for differences in skin thickness, the cartridge has an adjustable gauge to control injection depth. The syringe and electrodes are not visible to the participants. Once the device is placed on the recipient’s tissue site and activated by the operator, the electrodes are deployed into the tissue to the prescribed depth. The EP device first inserts the needle and the agent, then the electrode array which releases the electrical pulses into the tissue at the site of administration causing muscle contractions. 

b) The Integrated Applicator is a reusable hand held device that houses the single use cartridge. It deploys the electrodes and initiates the administration of the agent at the touch of the activation button. 

c) The pulse stimulator controls the administration sequences and generates the electroporation pulses. It is connected to the integrated applicator is connected to the pulse stimulator through an incorporated cable.

4. I assume from the patients comments that the needles were not initially visible. Was this one needle? Were there 4 electrode needles? 5 needles in total?

 Response: There was one needle and 4 electrodes all encased in the cartridge, as described above in (a).

5. The skin thickness was measured, but these were IM injections. How deep did the needles go? 

Response: Skin fold thickness of the upper deltoid region was measured using a caliper. In order to accommodate different skin thicknesses, the cartridge has an adjustable gauge to control injection depth. The range of this depth was 14-40 mm. A skin thickness of ≥ 40 mm was one of the exclusion criteria for the trial. This has been clarified in the text (Lines 135-137).

The range of the needle’s depth was 14-40 mm; those whose skin thickness was ≥ 40 mm were excluded from the trial.

6. Some positive patient comments included statements about reducing potential for error and contamination? Can I assume that the EP attachment is single use and disposable?

Response: Yes, the Application Cartridge (as described above) is single use and disposable.

7. Would it be possible to present a picture or diagram of the device for clarification?

Response: We thank the reviewer for this suggestion, and have inserted a picture of the device (line 107-108).

Comments from Reviewer 2

Summary: The analyses for this study are reasonably well done, but there is basically zero statistical power in these analyses. Hence, I question the utility of some of these analyses as I see the conclusions for some as misleading and potentially harmful.

Response. The parent study from which our data are drawn was designed to study the safety of IM/EP, and reported the vaccines to be safe, based on careful monitoring of study participants including medical history, physical exam, and lab assessments. Local and systemic signs and symptoms were collected for 7 days after each vaccination, and volunteers were monitored at the clinic on day 0 and on days 3 and 7 after each vaccination. A Protocol Safety Review Team supervised and monitored safety data on an ongoing basis, and an independent Safety Review Board reviewed interim safety data at 2 pre-determined time points. Although pause criteria were predefined in the study protocol, it was not necessary to use them.

The purpose of our secondary data analysis is to provide additional information to prospective vaccine trial participants by examining the data longitudinally within subjects. Our findings indicate that not all participants found the IM/EP acceptable, although the majority of the subjects did. Additionally, no participant ever reported “severe” or “very severe” levels of pain, the two highest pain categories offered as a possible response. This finding is supported by qualitative data and the findings of the parent study in terms of aggregate tolerability. While we appreciate the reviewer’s concerns, we do not feel that our results are misleading. Because the vaccine itself has been shown to be safe, encouraging otherwise reluctant persons to receive it or to agree to some form of electroporation as an administration system would not, we believe, be a harmful result.

1. (lines 144-145) I understand the allure of dichotomizing outcomes, but research suggests the loss of power after dichotomizing is large (e.g., https://doi.org/10.1002/pst.331). This seems like a really good opportunity to analyze the quantitative values. That will be a more powerful analysis and will provide a sense of how much the mean changes over time.

Response: We agree with the reviewer that dichotomizing an outcome results in a loss of power, and appreciate the suggestion of performing a more quantitative analysis. We chose not to do this for several reasons. First, because our outcome responses are ordinal and not continuous, using these as an outcome would have required ordinal logistic regression, the assumptions of which are difficult to prove, and the estimates from which are difficult to interpret. Second, our primary interest was in determining whether the electroporation process was perceived to be painful enough to contribute to the risk of loss-to-follow-up, or rather was considered acceptable enough to warrant subjects returning twice more after an initial exposure. We dichotomized responses between “uncomfortable” and “intense”, considering that few people consider any sort of injections to be comfortable or painless (we note that a limitation of our study is that we did not solicit reactions to a conventional injection for comparison). Finally, at no point did any person label pain as “severe” or “very severe;” we only observed responses of “none,” “light,” “uncomfortable,” or “intense,” making “intense” the evident choice for dichotomizing into too much pain. Had we observed a more nuanced reporting of the higher levels of pain, this may have been a more interesting option. 

2. (lines 167-174) I found this write up a little confusing. It sounds like logistic regression models are run at each time point and then a marginal model (using GEE) is also implemented. Are all time points included in the GEE-based model?

Response: We apologize for any confusion. All logistic regression models were performed using GEE techniques to account for within-person correlations. We have revised this part of the statistical analysis section to read as follows: (lines 189-200). 

Because our analyses included multiple observations per person, we examined the associations between time (across visits) and perception of pain with logistic regression models utilizing generalized estimating equations (GEE) techniques. GEE adjusts the variance of a repeated measures model in order to account for the within-person variability (due to multiple observations per person) in addition to the between-person variability.13 We first used univariate models to examine whether a participant’s perception of the pain associated with electroporation changed across vaccination visits. We analyzed a subject’s pain assessment at the time of the injection and at the time of electrical stimulation at visits 1, 2 and 3 (months 0, 1 and 2). This was repeated for the assessments of pain 10 minutes and 30 minutes post-injection, across all visits. We then repeated these analyses, adjusting for potential confounders including age, gender, BMI, and skin thickness. All analyses were conducted using Stata v14 (College Station, TX). Significance levels were set at an alpha of 0.05.

13Vittinghoff E, Glidden DV, Shiboski SC, McCulloch CE (2012). Regression methods in biostatistics: Linear, logistic, survival, and repeated measures models, 2nd edition. Springer: New York.

3. (Line 171) Please provide a methodological citation for GEE.

Response: Thank you for pointing out that we did not reference this statistical technique. See above for reference. GEE has now been referenced (l.192) in the text and added to the references. 

4. Please indicate the level of significance and the software package used in these analyses.

Response: Thank you for pointing out this oversight. We conducted all analyses using Stata v14 (College Station, TX), and significance levels were set at an alpha of 0.05. Both have been added to the text (ll.198-99). 

All analyses were conducted using Stata v14 (College Station, TX). Significance levels were set at an alpha of 0.05.

5. (lines 306-308) I think you need analyses specifically targeted at this. I may have missed something, but I don't think you have specifically tested this question in the manuscript. The modeling in this study has shown that perceptions of pain don't change over time, not that the levels of pain were acceptable and feasible. The proportion of participants reporting "too much pain" is presented without any sort of testing. I believe you will need to present some non-inferiority testing against a margin. Maybe in a non-trial setting 24% is not good and hinders a vaccination campaign.

Response: We appreciate the reviewer’s desire for statistical evidence that the trial is feasible or that the pain level was tolerable; however, it was not our intention to define tolerability, and we do not feel that doing so would necessarily be valid or useful. It is rather our intention to summarize the experience of our subjects as they went through three vaccinations. Each subject provided a total of 12 ratings throughout the study: four at each of three visits, for a total of 540 possible responses. Of these, pain was rated as “intense” only 16 times, with three individuals contributing seven of these ratings (and pain was never at any time rated as “severe” or “very severe”). Seventy-two percent of subjects rated the pain as acceptable at each of the 12 time points. We feel that these data provide useful information for use in recruiting for future trials. We also feel the perfect follow-up we observed, while it may not be proof of feasibility, does offer support for the feasibility of a trial utilizing electroporation.

6. (lines 308-310) I don't think the fact that everyone completed the series is robust evidence. There is the potential for compliance and reporting bias in trials, which makes this not the best justification.

Response: We agree with the reviewer that we cannot assume that acceptable toleration of the electroporation process “caused” the complete follow-up we observed. Other factors, such as interaction with and support of clinic staff, could certainly contribute to the subjects’ willingness to return for all visits. We have revised this statement (in the Discussion section, (ll. 366-69) as follows: 

While we cannot ascribe the complete follow-up we observed wholly to the acceptability of the electroporation process, we feel that perfect attendance does support the idea that the procedure itself was tolerable enough that subjects were not discouraged from returning to undergo it a second and third time.

7. (lines 313-315) Again, and apologies if I missed this, but I think you need to perform analyses specifically targeted at this question.

Response: Defining what constitutes a statistically significant proportion of patients whose pain lessons after they report intense pain at the time of electrical stimulation is outside the scope of our small study. We have revised this sentence in the Discussion section to make its descriptive nature more clear: (lines 373-376).

Seven subjects reported intense pain at the time of electrical stimulation (one patient reported intense pain at the time of electrical stimulation at each of the three visits; two more reported this at two visits each); in all 11 instances, however, the pain had lessened to an acceptable level by 10-minutes post injection.

8. (lines 322-325) While true that there were no major differences, the power to find a difference is basically nothing here. In Table 3, the 10 minutes post for visit 3 confidence intervals stretch from ~ 0.05 to ~6.00. At best, these models are essentially worthless and tell us nothing. At worst, conclusions are based off these results which basically have minimal chance of finding a difference. That's what is happening here. These conclusions are very misleading. Maybe if this study was a proof of concept study, then I would be more lenient, but you've stated that there are already other studies showing the same thing.

Response: GEE models are the only type of models that adjust standard errors for within-person variability, and were therefore necessary to examine longitudinal data. While the comparisons at 10 and 30 minutes post-injection likely are underpowered, the analysis of reactions at the time of electrical stimulation provide robust estimates with reasonable confidence intervals. We have modified our discussion of the results from GEE models to reflect these limitations (ll. 273-282) 

In univariate logistic regression models utilizing GEE techniques, we examined each subject’s reactions over time to the same time point; e.g., a subject’s response to electrical stimulation at Visit 1, Visit 2, and Visit 3. Likewise, we examined individual reactions across time at 10 minutes post IM/EP, and separately at 30 minutes post IM/EP. At the time of electrical stimulation, no significant difference was found in subject responses across visits. While the small size of our study does not rule out the possibility of having insufficient power to detect small differences, our results at this time point (where most “intense” reactions were reported) are supported by the fact the 72% of the cohort gave an acceptable rating at the time of electrical stimulation across all three visits. Therefore, a majority of subjects found the electrical stimulation itself acceptable, and this opinion did not change over time.

In evaluating reactions 10 minutes post IM/EP and 30 minutes post IM/EP across visits, we were unable to use the first visit as a reference as no subjects at that visit rated the pain at either 10 or 30 minutes post-injection to be unacceptable. This reduced our power to detect a difference between responses at Visit 3 using Visit 2 as a reference. While subjects appeared less likely at Visit 3 compared to Visit 2 to report an unacceptable level of pain at 10 minutes post IM/EP, this difference was not significant and confidence intervals were wide. At 30 minutes post IM/EP, no evidence was seen of a difference between reports of unacceptable pain at Visit 3 compared to Visit 2, but again, confidence intervals were quite wide for these comparisons. 

9. (lines 338-345) The survey or qualitative information you provide here may be the most interesting and novel about this manuscript. You might consider restructuring this paper and making it a qualitative manuscript.

Response. We thank the reviewer for this comment. The study from which we derived our data evaluated participants’ reactions to pain at different time points in aggregate. This does not take into account individual changes over time, but rather tests the overall group response at each time point. Given that we had longitudinal data, we saw an opportunity to do a more granular analysis by looking at reactions over time within individuals. Combining observations within a person generally reduces the variability in a model. Our data for the 45 subjects we included were complete. On the other hand, the qualitative data we collected were not complete, as each participant’s decision to add a comment at the end of the questionnaire was entirely optional. Some participants did not add any comments at any of the visits; others added comments at some but not all visits. Therefore, we feel that these data function best to support the conclusions of our quantitative analysis; this study does not truly represent a mixed methods approach.

Comments from Reviewer 3

The manuscript “Acceptability and tolerability of repeated intramuscular electroporation of Multiantigenic HIV (HIVMAG) DNA vaccine among healthy African participants in a phase 1

randomized controlled trial” by Mpendo et al describes the results from an ancillary study assessing the individual perceptions of tolerability over series of intramuscular (IM) electroporation (EP) vaccinations among healthy participants in three African countries, Uganda, Kenya and Rwanda. The parent study, also by Mpendo et al, was a multi-center, double-blind, randomized, placebo-controlled phase I trial, conducted between 2011-2013, assessing the safety and immunogenicity of heterologous prime-boost regimens using a multi-antigenic HIV (HIV-MAG) DNA vaccine or placebo as the prime vaccine at months 0, 1, and 2 with or without adjuvant (GENEVAX IL-12) and a booster vaccine using Ad35-GRIN/ENV or placebo. This ancillary study included 45 of the 75 subjects enrolled in the parent study (subjects who received EP immediately after the injection with the prime vaccine or placebo). The authors provide data and analyses for the research questions that are clearly summarized and represent important information for the vaccine field. I have no major comments or concerns regarding publication of this manuscript. I have minor recommendations (listed in order of appearance):

1. Affiliations: 

- Please list affiliations in numeric order

Response: Thank you for pointing this out. With the exception of the second author, we have reordered the affiliations in numeric order:

Juliet Mpendo1, Gaudensia Mutua2, Annet Nanvubya1, Omu Anzala2, Julien Nyonmbayire3, Etienne Karita3, Len Dally4, Drew Hannaman5, Patricia E. Fast6, Matt Price6, Frances Priddy6, Huub C. Gelderblom7, Nancy K. Hills8

Abstract:

1. Line 25: Please include “...Intramuscular (I.M.) electroporation” 

Response: Thank you for pointing this out. We have moved the full name of electroporation (as suggested) to line 21, where it is first mentioned, and have provided an abbreviation consistent with the remaining text. 

Intramuscular electroporation (IM/EP) is a vaccine delivery technique

2. Line 41: Should read “outcomes”? 

Response: Thank you for catching this, it has been corrected (line 41) 

Partnerships: 

3. Line 195: Is "system" duplicated. Response: 

Thank you for catching this error. The duplicated word has been removed. (Line 221) 

Results: 

4. Line 205: "...combined participants from all four groups...” Please revise. 

Response: This has been clarified as follows (ll.229-230):

We found no statistically significant difference in pain reaction at any time across the four groups (three vaccine and one placebo); therefore, we combined participants from all four groups to compare pain assessments at different times.

5. Line 208: 16 instances correspond to 12%. Please revise.

Response: Thank you for recalculating and catching this rounding error. This has been revised (line 240). 

6. Lines 230-232: From the text description you seem to only have 14 reports of "too much" pain vs the 16 reported in the table and text above. Response: We thank the reviewer for pointing out this discrepancy, which results from our not clearly distinguishing whether we were counting subjects who ever reported “too much” pain, or counting instances of reports (with multiple reports of too much pain by the same person at different times). We have added the following text in the second paragraph of the Results section. (Lines 230-238)

Each of the 45 subjects attended three vaccine sessions. At each of these sessions, the subjects’ reactions were measured at four separate time points relative to IM/EP. Therefore, each subject contributed three “pain ratings” at Time 0 (injection), one at each of the three vaccine visits, for a total of 135 total ratings at Time 0. Similarly, 135 ratings were collected at each of the following: Time 1 (electrical stimulation), Time 10 (10 minutes post IM/EP) and Time 30 (30 minutes post IM/EP). 

We have also added the following note to Table 2: 

Pain assessments at each of the three visits are counted by subject. However, in some cases the same subject reported pain at more than one time or visit. 

7. Lines 230-231: Please clarify if there were 5 participants who experienced "too much" pain after post-electroporation or there were fewer participants with repetitive events. 

- Lines 247-249: Your statement here is not aligned with you statement in lines 229-230 (on the first you say "in the majority of cases" and on the second you say "In every case"). Please include the remaining reported events of "too much pain" in lines 230-232 to ensure consistency. 

Response: Thank you for pointing out this confusion. We have revised (lines 246-54) to read:

A total of 16 responses of “too much” pain were reported by 11 subjects over the course of the study, with the majority of these reports at the time of electrical stimulation (11/16=68.5%). One person reported too much pain associated with electrical stimulation at each of the three visits; two others reported too much pain at two of the three electrical stimulations, and four reported too much pain at the time of electrical stimulation once only (two at Visit 1, one at Visit 2, and one at Visit 3). Three subjects who found the pain at the time of electrical stimulation to be acceptable did report too much pain 10 minutes after the injection, with one of these continuing to report pain 30 minutes after the injection. An additional subject reported too much pain 30 minutes after injection at the third visit only.

Acceptability of electroporation

8. You have only analyzed responses from the final visit, after the series of vaccinations had been completed. In a real life scenario it would be important to consider the perceptions after the first vaccination/EP since these may have an impact in retention. Please consider including this analysis and comparing the responses between the first and last visit. Also note that your study participants were under a very controlled research environment with possible extensive counselling which will differ in a real life scenario. 

Response: We thank the reviewer for pointing out the importance of evaluating the first as well as the last visit responses, and have revised this section of the paper (lines 346-357) as follows: 

Participants were asked at each visit if they felt that the electroporation procedure would be acceptable if it provided protection from serious disease. After the first vaccination, 44/45 of the participants responded positively, with one uncertain. After the second vaccination, 43/45 responded positively, with one each responding “no” and “uncertain.” After the third vaccination, 43/45 responded positively, with the remaining two responding “no.” Responses were similar when participants were asked if the electroporation procedure would be acceptable if it enhanced scientific knowledge about how best to administer/treat a disease. After the first vaccine, 44/45 responded positively, with one uncertain. After the last vaccine, 42/45 responded positively, with two responding “no” to both questions and one “uncertain.” The two subjects who responded “no”, as well as the subject who was “uncertain” about the second question only, had both reversed previously affirmative answers for both questions, although neither had reported “too much” pain at any time point or visit. 

Also note that your study participants were under a very controlled research environment with possible extensive counselling which will differ in a real life scenario.

The reviewer is correct that this should be noted; we have added the following text to the limitations section of the Discussion section of our paper (lines 421-24). 

In addition, we cannot be certain as to the generalizability of the study, since study participants participated in a controlled research environment with ample opportunities for questions, comments, and explanations of the process. In a real-world situation, it is unlikely that this degree of support would be available.

Discussion: 

9. Please comment on whether the grading of pain at the first visit may have been influenced by the fact that it was a complete new experience and the fear related to the procedure could have also played an important role. 

Response: We thank the reviewer for this suggestion. We have inserted the following into the Discussion section (ll. 381-84):

This was likely an emotional response to a new and “unknown” procedure, since the physical response, as rated quantitatively, was more often reported to be “intense” at some time point during visits 2 and 3 than at visit 1.

10. Lines 324-325: "... three electroporation procedures and one needle injection". It is one injection per electroporation. Please revise. 

Response: We apologize for this confusion. We were referring to the number of visits in the entire series (four). We have clarified our meaning with the following revision in the text (ll. 387-91)

No major differences in pain were reported within individuals across visits, and all participants returned as required and completed the entire vaccination series that consisted of a total of four visits: three visits at which the electroporation procedure was used to administer the vaccine, and one visit (the last) at which needle injection was used rather than electroporation.

---

## [Decision Letter · Decision Letter 1]

17 Dec 2019

PONE-D-19-22032R1

Acceptability and tolerability of repeated intramuscular electroporation of Multi-antigenic HIV (HIVMAG) DNA vaccine among healthy African participants in a phase 1 randomized controlled trial.

PLOS ONE

Dear Dr MPENDO,

Thank you for submitting your manuscript to PLOS ONE. After careful consideration, we feel that it has merit but does not fully meet PLOS ONE’s publication criteria as it currently stands. Therefore, we invite you to submit a revised version of the manuscript that addresses the points raised during the review process.

We would appreciate receiving your revised manuscript by Jan 31 2020 11:59PM. To enhance the reproducibility of your results, we recommend that if applicable you deposit your laboratory protocols in protocols.io, where a protocol can be assigned its own identifier (DOI) such that it can be cited independently in the future. For instructions see: http://journals.plos.org/plosone/s/submission-guidelines#loc-laboratory-protocols

We look forward to receiving your revised manuscript.

Kind regards,

David Joseph Diemert, M.D.

Academic Editor

PLOS ONE

Reviewers' comments:

Reviewer's Responses to Questions

**Comments to the Author**

1. If the authors have adequately addressed your comments raised in a previous round of review and you feel that this manuscript is now acceptable for publication, you may indicate that here to bypass the “Comments to the Author” section, enter your conflict of interest statement in the “Confidential to Editor” section, and submit your "Accept" recommendation.

Reviewer #1: All comments have been addressed

Reviewer #2: (No Response)

2. Is the manuscript technically sound, and do the data support the conclusions?

Reviewer #1: Yes

Reviewer #2: No

3. Has the statistical analysis been performed appropriately and rigorously? 

Reviewer #1: Yes

Reviewer #2: No

4. Have the authors made all data underlying the findings in their manuscript fully available?

Reviewer #1: Yes

Reviewer #2: Yes

5. Is the manuscript presented in an intelligible fashion and written in standard English?

Reviewer #1: Yes

Reviewer #2: Yes

6. Review Comments to the Author

Reviewer #1: (No Response)

Reviewer #2: Thank you for your thorough and thoughtful consideration of my comments and for clarifying the statistical methods.

That said, I was somewhat perplexed by some of your responses to my comments. My goal as a reviewer, especially a statistical reviewer, is to ensure criteria 3 and 4 of the publication criteria are met. My desire for statistical evidence is based criterion #4, "conclusions are presented in an appropriate fashion and are supported by the data." I am only responding to the statements that you have made in the manuscript. In order to be in accordance with the publication criteria, I am asking for evidence to be presented which supports those claims. My "desire" for empirical evidence is based on the journal's desire for empirical evidence. Of course, I may have misread or misinterpreted certain statements and you should feel free to correct me, but that's where I am coming from.

Second, a technical comment: for GEE models to be robust, there needs to be around 20 groups per condition (https://doi.org/10.2105/AJPH.94.3.423). That's because marginal models are trying to make conclusions at the aggregate level and, when there are not many groups (in your case, people), there's been simulation studies that show the results became less robust once the number of groups starts dropping below that level.

In addition, regarding the modeling, you stated in your responses that, "…our primary interest was in determining whether the electroporation process was perceived to be painful enough to contribute to the risk of loss-to-follow-up, or rather was considered acceptable enough to warrant subjects returning twice more after an initial exposure." This sounds to me like pain levels are the explanatory variable and that LTFU/retention is the outcome. If that is the case, then the models presented need to be changed to reflect this.

Regarding the Likert-like scale for pain, there is large disagreement on whether such a scale can/should be treated as continuous. I am still not comfortable with this, but there's not much that can be done at this stage other than to say to please try to implement a more nuanced pain scale in future studies.

Though, putting the modeling aside, as I read through your responses and revisions, I still maintain that you are making this manuscript much harder and complex than it needs to be. As I see it, you have two main points: (1) almost all patients rated this procedure as acceptable and (2) all the patients were retained. Those are important points and I agree, even with a small sample size, that these will support your end goal of showing that the acceptability and tolerability of this procedure. But, the modeling convolutes the manuscript and I don't think it needs to be there. Further, again pulling from your comments, I think this is what you really want: "It is rather our intention to summarize the experience of our subjects as they went through three vaccinations."

7. PLOS authors have the option to publish the peer review history of their article (what does this mean?). If published, this will include your full peer review and any attached files.

Reviewer #1: No

Reviewer #2: No

---

## [Author Response · Author response to Decision Letter 1]

5 Feb 2020

Dear Dr. Diemert,

Thank you for being willing to solicit another opinion on our manuscript. We very much appreciate this opportunity, and have responded to the additional reviewer comments below, as you requested. Thank you again for your consideration of our concerns.

Author issues with prior review:

As the statistician who designed and conducted this analysis, I stand by our methodological decisions and feel that the reviewer has misunderstood our analysis and our explanations in response to stated concerns about it. Specifically, the reviewer disagrees with our use of GEE techniques to account for the correlations created by inclusion of multiple measurements per person. Although a prior study authored by Dr. Mpendo had looked at subject responses to IM/EP in aggregate, she wished to look at individual responses over time, requiring repeated measures analysis.

We have found the reviewer’s concern that the paper is misleading and potentially harmful to be truly puzzling, and have responded to this concern to the best of our ability. Although the reviewer has asked us to remove the analysis which is at the heart of this paper, this would repeat analyses already published, and be counter to the purpose of the paper. I do not agree with the Reviewer’s belief that GEE is inappropriate here; GEE merely adjusts the standard errors in a model to include the within person variability that is generated when multiple observations are made on the same person. (And then averages over all observations.)

We thank the editors for allowing further consideration of our manuscript, despite what the prior reviewer feels are fatal flaws. Should you decide not to publish the paper, we will accept your decision and withdraw it.

Respectfully yours,

Drs. Juliet Mpendo, Nancy Hills, Matt Price and Pat Fast

Second round comments from reviewer and our responses:

Reviewer #2: Thank you for your thorough and thoughtful consideration of my comments and for clarifying the statistical methods.

That said, I was somewhat perplexed by some of your responses to my comments. My goal as a reviewer, especially a statistical reviewer, is to ensure criteria 3 and 4 of the publication criteria are met. My desire for statistical evidence is based criterion #4, "conclusions are presented in an appropriate fashion and are supported by the data." I am only responding to the statements that you have made in the manuscript. In order to be in accordance with the publication criteria, I am asking for evidence to be presented which supports those claims. My "desire" for empirical evidence is based on the journal's desire for empirical evidence. Of course, I may have misread or misinterpreted certain statements and you should feel free to correct me, but that's where I am coming from.

Response: We feel that we have responded to the reviewer’s concerns to the best of our ability.

Second, a technical comment: for GEE models to be robust, there needs to be around 20 groups per condition (https://doi.org/10.2105/AJPH.94.3.423). That's because marginal models are trying to make conclusions at the aggregate level and, when there are not many groups (in your case, people), there's been simulation studies that show the results became less robust once the number of groups starts dropping below that level.

Response: The subject of the article to which the reviewer refers is actually group-randomized trials; i.e., the group is the unit of analysis. This presents different problems than we address here; our purpose in utilizing GEE is simply to adjust the standard errors in the analysis to account for the added within-person variability created when multiple measurements are made on the same person. While our small sample may not represent the most ideal characteristics for GEE analysis, it certainly does not preclude its use. 

In addition, regarding the modeling, you stated in your responses that, "…our primary interest was in determining whether the electroporation process was perceived to be painful enough to contribute to the risk of loss-to-follow-up, or rather was considered acceptable enough to warrant subjects returning twice more after an initial exposure." This sounds to me like pain levels are the explanatory variable and that LTFU/retention is the outcome. If that is the case, then the models presented need to be changed to reflect this.

Response: Our primary purpose, as stated, was to examine time as a predictor of reported pain from the electroporation procedure. Since follow-up in our study is perfect, it would not be possible to use retention as an outcome. We have tried to make clear that we believe that the evidence we found supports acceptability of the procedure and retention in the study, and not that it causes it or proves it.

Regarding the Likert-like scale for pain, there is large disagreement on whether such a scale can/should be treated as continuous. I am still not comfortable with this, but there's not much that can be done at this stage other than to say to please try to implement a more nuanced pain scale in future studies.

Response: We do not agree with reviewer that a Likert-scale response should be viewed as continuous, for the reasons stated in our prior response.

Though, putting the modeling aside, as I read through your responses and revisions, I still maintain that you are making this manuscript much harder and complex than it needs to be. As I see it, you have two main points: (1) almost all patients rated this procedure as acceptable and (2) all the patients were retained. Those are important points and I agree, even with a small sample size, that these will support your end goal of showing that the acceptability and tolerability of this procedure. But, the modeling convolutes the manuscript and I don't think it needs to be there. 

Response: Acceptability data in aggregate have already been published, as we pointed out. We do not view a model that merely accounts for the correlation between multiple observations on the same person to be complex or convoluted.

Further, again pulling from your comments, I think this is what you really want: "It is rather our intention to summarize the experience of our subjects as they went through three vaccinations."

Response: We feel that our methodology has been appropriate to address this issue which necessarily must account for repeated observations per person.

---

## [Decision Letter · Decision Letter 2]

23 Mar 2020

PONE-D-19-22032R2

Acceptability and tolerability of repeated intramuscular electroporation of Multi-antigenic HIV (HIVMAG) DNA vaccine among healthy African participants in a phase 1 randomized controlled trial.

PLOS ONE

Dear Dr MPENDO,

Thank you for submitting your manuscript to PLOS ONE. After careful re-consideration and review by an additional statistician, we feel that it has merit but does not fully meet PLOS ONE’s publication criteria as it currently stands. Therefore, we invite you to submit a revised version of the manuscript that addresses the points raised during the most recent review process.

We would appreciate receiving your revised manuscript by May 07 2020 11:59PM. To enhance the reproducibility of your results, we recommend that if applicable you deposit your laboratory protocols in protocols.io, where a protocol can be assigned its own identifier (DOI) such that it can be cited independently in the future. For instructions see: http://journals.plos.org/plosone/s/submission-guidelines#loc-laboratory-protocols

We look forward to receiving your revised manuscript.

Kind regards,

David Joseph Diemert, M.D.

Academic Editor

PLOS ONE

Reviewers' comments:

Reviewer's Responses to Questions

**Comments to the Author**

1. If the authors have adequately addressed your comments raised in a previous round of review and you feel that this manuscript is now acceptable for publication, you may indicate that here to bypass the “Comments to the Author” section, enter your conflict of interest statement in the “Confidential to Editor” section, and submit your "Accept" recommendation.

Reviewer #4: (No Response)

2. Is the manuscript technically sound, and do the data support the conclusions?

Reviewer #4: Yes

3. Has the statistical analysis been performed appropriately and rigorously? 

Reviewer #4: Yes

4. Have the authors made all data underlying the findings in their manuscript fully available?

Reviewer #4: Yes

5. Is the manuscript presented in an intelligible fashion and written in standard English?

Reviewer #4: Yes

6. Review Comments to the Author

Reviewer #4: This is an important study carrying out a repeated measures analysis looking at tolerability and acceptability of EP vaccinations in health adults. The paper is well written and clear. They are some minor comments worth adding or clarifying in the manuscript.

Stats section mentions reporting mean(SD) for normally distributed data, median(IQR) have also been presented so this should be included as an additional text. As well as stating freq(%) were also presented for categorical data.

Line 181 "All responses were recorded by administrators on paper questionnaires, and later entered into an Excel data base.", it would be good to state who entered the data, i.e independent of the study, or part of the study group, was this data double entered for example to make sure there were no errors, i.e. to ensure data quality.

A GEE techniques used, the link function used should be stated.

line 199 "We then repeated these analyses, adjusting for potential confounders 200 including age, gender, BMI, and skin thickness.2 Were these pre-specified?

7. PLOS authors have the option to publish the peer review history of their article (what does this mean?). If published, this will include your full peer review and any attached files.

Reviewer #4: No

---

## [Author Response · Author response to Decision Letter 2]

24 Apr 2020

20th April 2020

David Joseph Diemert

Academic Editor

PLOS ONE

Dear Dr. Diemert,

Thank you so much for sending our manuscript to an additional statistical reviewer. We very much appreciate your consideration and willingness to address our concerns. We carefully considered the additional comments from Reviewer #4, and have addressed them to the best of our ability. 

The following is a point-by-point explanation of how we have addressed the concerns and revised our manuscript. The line number(s) on the revised manuscript with Track changes have been specified to show the location of the text that has been changed/added in response to the reviewer comments. We have not re-edited our responses to previous reviewers, but we have added (in red) the page numbers as they occur in our most recently edited manuscript.

Again, thank you so much for your time on and attention to our manuscript. We would be very proud to have it appear in PLOS ONE.

Sincerely, 

Juliet

Comments from Reviewer

1. Reviewer #4: This is an important study carrying out a repeated measures analysis looking at tolerability and acceptability of EP vaccinations in health adults. The paper is well written and clear. 

We thank the reviewer for taking the time to review our manuscript, and for this favorable opinion.

There are some minor comments worth adding or clarifying in the manuscript. Stats section mentions reporting mean (SD) for normally distributed data, median (IQR) have also been presented so this should be included as an additional text. As well as stating freq (%) were also presented for categorical data.

Response: Thank you for pointing out this omission. The first sentences of the Statistical Analysis section have been edited as follows (ll.189-92): 

Demographic characteristics of the cohort were summarized using means and standard deviations for normally distributed continuous variables and medians and IQRs for continuous variables that were not normally distributed. Categorical variables were summarized using frequencies and percentages. 

2. Line 181 "All responses were recorded by administrators on paper questionnaires, and later entered into an Excel data base.", it would be good to state who entered the data, i.e. independent of the study, or part of the study group, was this data double entered for example to make sure there were no errors, i.e. to ensure data quality.

Response: Although the data were single-entered by the nurses who were part of the study team, the data manager verified all data entries against the paper source questionnaires. We have clarified this in the text as follows (ll.183-86): 

All responses were recorded by the study nurses on paper questionnaires, and later entered into an Excel database by data entry officers who were part of the study team. The data manager was responsible for verifying the data entered against the paper source.

3. A GEE techniques used, the link function used should be stated.

Response: We have added the following information to the text (ll.200-01): 

We ran logistic GEE models, with a logit link and an exchangeable correlation structure with robust standard errors.

4. line 199 "We then repeated these analyses, adjusting for potential confounders including age, gender, BMI, and skin thickness.2 Were these pre-specified?

Response: Our study involved secondary data analysis of a clinical trial (the number of which we have now included in the Methods section, l.81): 

The parent study, a multi-center, double-blind, randomized, placebo-controlled phase I trial (ClinicalTrials.gov NCT01496989) designed to evaluate the safety and immunogenicity of heterologous prime-boost regimens, was conducted from December 2011 to March 2013 in urban settings in Uganda, Kenya and Rwanda(7).

Response, continued. Therefore, these data were collected for the parent trial and the confounders we examined were not pre-specified for the primary analysis, which was focused on safety and immunogenicity. Skin thickness was tested for the parent trial in order that settings to the electroporation apparatus could be adjusted accordingly: [Protocol, p.23, section 5.2.4. Exclusion Criteria #25. “Skin and subcutaneous tissue thickness > 40 mm as assessed by skin pinch test in either deltoid region”]

We hypothesized that skin thickness and BMI could possibly/likely be associated with the discomfort a participant felt from the injections, and therefore adjusted for them in our analyses. We also adjusted for age and gender as factors potentially associated with a participant’s response to the injections.

[Previous comments and responses appear on the following pages.]

 

Prior comments and responses: NOTE: We have added page numbers corresponding to our current manuscript to previous responses in red.

Mpendo, et al, have presented a subset of data from a previously performed clinical trial assessing the acceptability of intramuscular electroporation as a delivery method for DNA vaccination. Researchers continue to explore methods to enhance the immunogenicity of DNA vaccines and electroporation is increasingly touted as a potentially viable method. While it may be determined that electroporation sufficiently increases immunologic responses, it will not be an effective vaccination method if patients are unwilling to undergo the procedure. Results such as these will be viewed as important in determining whether or not to continue to explore this method of delivery.

The authors have done a fine job presenting their data. The manuscript is well written and clear. I have only minor questions or comments:

1. The parent study was 2011-2013. That is a significant delay in publishing these results. Can the authors comment on why this is now seen as important enough to publish? 

Response: Novel methods of HIV vaccine administration that will provide good immunogenicity still need to be explored. Although the parent study was conducted from 2011-2013, the website ClinicalTrials.gov currently reports a number of ongoing and recruiting trials utilizing electroporation DNA vaccine delivery. We therefore believe that our data can still provide valuable information, especially for recruiting purposes, in helping to assure potential trial subjects of the acceptability of the process. Because we had the opportunity to assess the reactions to the electroporation process in individuals over the course of time, this can be offered as support that the process was not only acceptable to subjects on a one-time basis, but acceptable enough to guarantee their return for multiple procedures. 

2. There were 75 people in the original study. They only included 45 in this study (the three groups who received multiple EP injections). They note in the Discussion that these data were not collected on patients receiving the vaccine by standard administration. Was this same information gathered on any others in the study and just not included in this analysis? If so, would the results change if they were included? 

Response: Data on pain/acceptability were collected on all 75 participants upon receipt of EP. The parent study assessed pain/acceptability in aggregate at each time point. They found no statistically significant difference in proportions of volunteers who rated the procedure uncomfortable or intense at any time point among the five vaccine and one placebo groups.1 Our analysis is unique in that we looked at responses to the EP procedure within individuals across the series of vaccines. This allowed us to ascertain whether reactions to repeated injections over time were acceptable within each individual. We therefore excluded from our analyses the 30 participants who only received EP at one time point, as they could not add any information to our longitudinal analysis. A comparison of these 30 to the remaining 45 at the first visit revealed no differences in response between the two groups (EP once only vs. EP multiple times). We have clarified this in the text and added information (lines 89-94): 92-96

…we included in this analysis only participants from the three groups who received multiple electroporation vaccinations (excluding those without longitudinal data), giving us a sample size of 45. A comparison of the reactions of the 30 excluded subjects to the remaining 45 at the first visit revealed no differences in response between the two groups.

1Mpendo J, Mutua G, Nyombayire J, Ingabire R, Nanvubya A, Anzala O, et al. A Phase I Double Blind, Placebo-Controlled, Randomized Study of the Safety and Immunogenicity of Electroporated HIV DNA with or without Interleukin 12 in Prime-Boost Combinations with an Ad35 HIV Vaccine in Healthy HIV-Seronegative African Adults. PloS one. 2015;10(8):e0134287.

3. Please clarify: When the device was held against the participant’s arm for a second or 2, were needles inserted into the arm at that point or were they triggered (“fired”) by pressing the activation button? 

Response: The insertion of the needle is triggered by pressing the activation button. We thank the reviewer for pointing out the need for further clarification on the EP process. We have briefly described the TriGrid delivery system, which is comprised of three components, to more clearly illustrate the procedure. The text below has been inserted into the manuscript (lines 109-125) 112-128

a) The Application Cartridge is a sterile, single-use component that houses the agent to be delivered (in a standard syringe) and a TriGrid electrode array for the EP procedure. The cartridge is the only subject contact of the system. It encloses 4 electrodes. Prior to administration procedure, the syringe containing the agent is loaded into the application cartridge. Once loaded with the agent, the cartridge is attached to the integrated applicator for administration to the recipient. In order to cater for differences in skin thickness, the cartridge has an adjustable gauge to control injection depth. The syringe and electrodes are not visible to the participants. Once the device is placed on the recipient’s tissue site and activated by the operator, the electrodes are deployed into the tissue to the prescribed depth. The EP device first inserts the needle and the agent, then the electrode array which releases the electrical pulses into the tissue at the site of administration causing muscle contractions. 

b) The Integrated Applicator is a reusable hand held device that houses the single use cartridge. It deploys the electrodes and initiates the administration of the agent at the touch of the activation button. 

c) The pulse stimulator controls the administration sequences and generates the electroporation pulses. It is connected to the integrated applicator is connected to the pulse stimulator through an incorporated cable.

4. I assume from the patients comments that the needles were not initially visible. Was this one needle? Were there 4 electrode needles? 5 needles in total?

 Response: There was one needle and 4 electrodes all encased in the cartridge, as described above in (a).

5. The skin thickness was measured, but these were IM injections. How deep did the needles go? 

Response: Skin fold thickness of the upper deltoid region was measured using a caliper. In order to accommodate different skin thicknesses, the cartridge has an adjustable gauge to control injection depth. The range of this depth was 14-40 mm. A skin thickness of ≥ 40 mm was one of the exclusion criteria for the trial. This has been clarified in the text (Lines 135-137). 138-139

The range of the needle’s depth was 14-40 mm; those whose skin thickness was ≥ 40 mm were excluded from the trial.

6. Some positive patient comments included statements about reducing potential for error and contamination? Can I assume that the EP attachment is single use and disposable?

Response: Yes, the Application Cartridge (as described above) is single use and disposable.

7. Would it be possible to present a picture or diagram of the device for clarification?

Response: We thank the reviewer for this suggestion, and have inserted a picture of the device (line 107-108). 109-111

Comments from Reviewer 2

Summary: The analyses for this study are reasonably well done, but there is basically zero statistical power in these analyses. Hence, I question the utility of some of these analyses as I see the conclusions for some as misleading and potentially harmful.

Response. The parent study from which our data are drawn was designed to study the safety of IM/EP, and reported the vaccines to be safe, based on careful monitoring of study participants including medical history, physical exam, and lab assessments. Local and systemic signs and symptoms were collected for 7 days after each vaccination, and volunteers were monitored at the clinic on day 0 and on days 3 and 7 after each vaccination. A Protocol Safety Review Team supervised and monitored safety data on an ongoing basis, and an independent Safety Review Board reviewed interim safety data at 2 pre-determined time points. Although pause criteria were predefined in the study protocol, it was not necessary to use them.

The purpose of our secondary data analysis is to provide additional information to prospective vaccine trial participants by examining the data longitudinally within subjects. Our findings indicate that not all participants found the IM/EP acceptable, although the majority of the subjects did. Additionally, no participant ever reported “severe” or “very severe” levels of pain, the two highest pain categories offered as a possible response. This finding is supported by qualitative data and the findings of the parent study in terms of aggregate tolerability. While we appreciate the reviewer’s concerns, we do not feel that our results are misleading. Because the vaccine itself has been shown to be safe, encouraging otherwise reluctant persons to receive it or to agree to some form of electroporation as an administration system would not, we believe, be a harmful result.

1. (lines 144-145) I understand the allure of dichotomizing outcomes, but research suggests the loss of power after dichotomizing is large (e.g., https://doi.org/10.1002/pst.331). This seems like a really good opportunity to analyze the quantitative values. That will be a more powerful analysis and will provide a sense of how much the mean changes over time.

Response: We agree with the reviewer that dichotomizing an outcome results in a loss of power, and appreciate the suggestion of performing a more quantitative analysis. We chose not to do this for several reasons. First, because our outcome responses are ordinal and not continuous, using these as an outcome would have required ordinal logistic regression, the assumptions of which are difficult to prove, and the estimates from which are difficult to interpret. Second, our primary interest was in determining whether the electroporation process was perceived to be painful enough to contribute to the risk of loss-to-follow-up, or rather was considered acceptable enough to warrant subjects returning twice more after an initial exposure. We dichotomized responses between “uncomfortable” and “intense”, considering that few people consider any sort of injections to be comfortable or painless (we note that a limitation of our study is that we did not solicit reactions to a conventional injection for comparison). Finally, at no point did any person label pain as “severe” or “very severe;” we only observed responses of “none,” “light,” “uncomfortable,” or “intense,” making “intense” the evident choice for dichotomizing into too much pain. Had we observed a more nuanced reporting of the higher levels of pain, this may have been a more interesting option. 

2. (lines 167-174) I found this write up a little confusing. It sounds like logistic regression models are run at each time point and then a marginal model (using GEE) is also implemented. Are all time points included in the GEE-based model?

Response: We apologize for any confusion. All logistic regression models were performed using GEE techniques to account for within-person correlations. We have revised this part of the statistical analysis section to read as follows: (lines 189-200). 196-209

Because our analyses included multiple observations per person, we examined the associations between time (across visits) and perception of pain with logistic regression models utilizing generalized estimating equations (GEE) techniques. GEE adjusts the variance of a repeated measures model in order to account for the within-person variability (due to multiple observations per person) in addition to the between-person variability.13 We first used univariate models to examine whether a participant’s perception of the pain associated with electroporation changed across vaccination visits. We analyzed a subject’s pain assessment at the time of the injection and at the time of electrical stimulation at visits 1, 2 and 3 (months 0, 1 and 2). This was repeated for the assessments of pain 10 minutes and 30 minutes post-injection, across all visits. We then repeated these analyses, adjusting for potential confounders including age, gender, BMI, and skin thickness. All analyses were conducted using Stata v14 (College Station, TX). Significance levels were set at an alpha of 0.05.

13Vittinghoff E, Glidden DV, Shiboski SC, McCulloch CE (2012). Regression methods in biostatistics: Linear, logistic, survival, and repeated measures models, 2nd edition. Springer: New York.

3. (Line 171) Please provide a methodological citation for GEE.

Response: Thank you for pointing out that we did not reference this statistical technique. See above for reference. GEE has now been referenced (l.192) 200 in the text and added to the references. 

4. Please indicate the level of significance and the software package used in these analyses.

Response: Thank you for pointing out this oversight. We conducted all analyses using Stata v14 (College Station, TX), and significance levels were set at an alpha of 0.05. Both have been added to the text (ll.198-99). 208-209

All analyses were conducted using Stata v14 (College Station, TX). Significance levels were set at an alpha of 0.05.

5. (lines 306-308) I think you need analyses specifically targeted at this. I may have missed something, but I don't think you have specifically tested this question in the manuscript. The modeling in this study has shown that perceptions of pain don't change over time, not that the levels of pain were acceptable and feasible. The proportion of participants reporting "too much pain" is presented without any sort of testing. I believe you will need to present some non-inferiority testing against a margin. Maybe in a non-trial setting 24% is not good and hinders a vaccination campaign.

Response: We appreciate the reviewer’s desire for statistical evidence that the trial is feasible or that the pain level was tolerable; however, it was not our intention to define tolerability, and we do not feel that doing so would necessarily be valid or useful. It is rather our intention to summarize the experience of our subjects as they went through three vaccinations. Each subject provided a total of 12 ratings throughout the study: four at each of three visits, for a total of 540 possible responses. Of these, pain was rated as “intense” only 16 times, with three individuals contributing seven of these ratings (and pain was never at any time rated as “severe” or “very severe”). Seventy-two percent of subjects rated the pain as acceptable at each of the 12 time points. We feel that these data provide useful information for use in recruiting for future trials. We also feel the perfect follow-up we observed, while it may not be proof of feasibility, does offer support for the feasibility of a trial utilizing electroporation.

6. (lines 308-310) I don't think the fact that everyone completed the series is robust evidence. There is the potential for compliance and reporting bias in trials, which makes this not the best justification.

Response: We agree with the reviewer that we cannot assume that acceptable toleration of the electroporation process “caused” the complete follow-up we observed. Other factors, such as interaction with and support of clinic staff, could certainly contribute to the subjects’ willingness to return for all visits. We have revised this statement (in the Discussion section, (ll. 366-69) as follows: 399-402

While we cannot ascribe the complete follow-up we observed wholly to the acceptability of the electroporation process, we feel that perfect attendance does support the idea that the procedure itself was tolerable enough that subjects were not discouraged from returning to undergo it a second and third time.

7. (lines 313-315) Again, and apologies if I missed this, but I think you need to perform analyses specifically targeted at this question.

Response: Defining what constitutes a statistically significant proportion of patients whose pain lessons after they report intense pain at the time of electrical stimulation is outside the scope of our small study. We have revised this sentence in the Discussion section to make its descriptive nature more clear: (lines 373-376). 408-412

Seven subjects reported intense pain at the time of electrical stimulation (one patient reported intense pain at the time of electrical stimulation at each of the three visits; two more reported this at two visits each); in all 11 instances, however, the pain had lessened to an acceptable level by 10-minutes post injection.

8. (lines 322-325) While true that there were no major differences, the power to find a difference is basically nothing here. In Table 3, the 10 minutes post for visit 3 confidence intervals stretch from ~ 0.05 to ~6.00. At best, these models are essentially worthless and tell us nothing. At worst, conclusions are based off these results which basically have minimal chance of finding a difference. That's what is happening here. These conclusions are very misleading. Maybe if this study was a proof of concept study, then I would be more lenient, but you've stated that there are already other studies showing the same thing.

Response: GEE models are the only type of models that adjust standard errors for within-person variability, and were therefore necessary to examine longitudinal data. While the comparisons at 10 and 30 minutes post-injection likely are underpowered, the analysis of reactions at the time of electrical stimulation provide robust estimates with reasonable confidence intervals. We have modified our discussion of the results from GEE models to reflect these limitations (ll. 273-282) 287-306

In univariate logistic regression models utilizing GEE techniques, we examined each subject’s reactions over time to the same time point; e.g., a subject’s response to electrical stimulation at Visit 1, Visit 2, and Visit 3. Likewise, we examined individual reactions across time at 10 minutes post IM/EP, and separately at 30 minutes post IM/EP. At the time of electrical stimulation, no significant difference was found in subject responses across visits. While the small size of our study does not rule out the possibility of having insufficient power to detect small differences, our results at this time point (where most “intense” reactions were reported) are supported by the fact the 72% of the cohort gave an acceptable rating at the time of electrical stimulation across all three visits. Therefore, a majority of subjects found the electrical stimulation itself acceptable, and this opinion did not change over time.

In evaluating reactions 10 minutes post IM/EP and 30 minutes post IM/EP across visits, we were unable to use the first visit as a reference as no subjects at that visit rated the pain at either 10 or 30 minutes post-injection to be unacceptable. This reduced our power to detect a difference between responses at Visit 3 using Visit 2 as a reference. While subjects appeared less likely at Visit 3 compared to Visit 2 to report an unacceptable level of pain at 10 minutes post IM/EP, this difference was not significant and confidence intervals were wide. At 30 minutes post IM/EP, no evidence was seen of a difference between reports of unacceptable pain at Visit 3 compared to Visit 2, but again, confidence intervals were quite wide for these comparisons. 

9. (lines 338-345) The survey or qualitative information you provide here may be the most interesting and novel about this manuscript. You might consider restructuring this paper and making it a qualitative manuscript.

Response. We thank the reviewer for this comment. The study from which we derived our data evaluated participants’ reactions to pain at different time points in aggregate. This does not take into account individual changes over time, but rather tests the overall group response at each time point. Given that we had longitudinal data, we saw an opportunity to do a more granular analysis by looking at reactions over time within individuals. Combining observations within a person generally reduces the variability in a model. Our data for the 45 subjects we included were complete. On the other hand, the qualitative data we collected were not complete, as each participant’s decision to add a comment at the end of the questionnaire was entirely optional. Some participants did not add any comments at any of the visits; others added comments at some but not all visits. Therefore, we feel that these data function best to support the conclusions of our quantitative analysis; this study does not truly represent a mixed methods approach.

Comments from Reviewer 3

The manuscript “Acceptability and tolerability of repeated intramuscular electroporation of Multiantigenic HIV (HIVMAG) DNA vaccine among healthy African participants in a phase 1

randomized controlled trial” by Mpendo et al describes the results from an ancillary study assessing the individual perceptions of tolerability over series of intramuscular (IM) electroporation (EP) vaccinations among healthy participants in three African countries, Uganda, Kenya and Rwanda. The parent study, also by Mpendo et al, was a multi-center, double-blind, randomized, placebo-controlled phase I trial, conducted between 2011-2013, assessing the safety and immunogenicity of heterologous prime-boost regimens using a multi-antigenic HIV (HIV-MAG) DNA vaccine or placebo as the prime vaccine at months 0, 1, and 2 with or without adjuvant (GENEVAX IL-12) and a booster vaccine using Ad35-GRIN/ENV or placebo. This ancillary study included 45 of the 75 subjects enrolled in the parent study (subjects who received EP immediately after the injection with the prime vaccine or placebo). The authors provide data and analyses for the research questions that are clearly summarized and represent important information for the vaccine field. I have no major comments or concerns regarding publication of this manuscript. I have minor recommendations (listed in order of appearance):

1. Affiliations: 

- Please list affiliations in numeric order

Response: Thank you for pointing this out. With the exception of the second author, we have reordered the affiliations in numeric order:

Juliet Mpendo1, Gaudensia Mutua2, Annet Nanvubya1, Omu Anzala2, Julien Nyonmbayire3, Etienne Karita3, Len Dally4, Drew Hannaman5, Patricia E. Fast6, Matt Price6, Frances Priddy6, Huub C. Gelderblom7, Nancy K. Hills8

Abstract:

1. Line 25: Please include “...Intramuscular (I.M.) electroporation” 

Response: Thank you for pointing this out. We have moved the full name of electroporation (as suggested) to line 21, where it is first mentioned, and have provided an abbreviation consistent with the remaining text. 

Intramuscular electroporation (IM/EP) is a vaccine delivery technique

2. Line 41: Should read “outcomes”? 

Response: Thank you for catching this, it has been corrected (line 41) 

Partnerships: 

3. Line 195: Is "system" duplicated. Response: 

Thank you for catching this error. The duplicated word has been removed. (Line 221) 231

Results: 

4. Line 205: "...combined participants from all four groups...” Please revise. 

Response: This has been clarified as follows (ll.229-230): 240-242

We found no statistically significant difference in pain reaction at any time across the four groups (three vaccine and one placebo); therefore, we combined participants from all four groups to compare pain assessments at different times.

5. Line 208: 16 instances correspond to 12%. Please revise.

Response: Thank you for recalculating and catching this rounding error. This has been revised (line 240). 250

6. Lines 230-232: From the text description you seem to only have 14 reports of "too much" pain vs the 16 reported in the table and text above. Response: We thank the reviewer for pointing out this discrepancy, which results from our not clearly distinguishing whether we were counting subjects who ever reported “too much” pain, or counting instances of reports (with multiple reports of too much pain by the same person at different times). We have added the following text in the second paragraph of the Results section. (Lines 230-238) 242-247

Each of the 45 subjects attended three vaccine sessions. At each of these sessions, the subjects’ reactions were measured at four separate time points relative to IM/EP. Therefore, each subject contributed three “pain ratings” at Time 0 (injection), one at each of the three vaccine visits, for a total of 135 total ratings at Time 0. Similarly, 135 ratings were collected at each of the following: Time 1 (electrical stimulation), Time 10 (10 minutes post IM/EP) and Time 30 (30 minutes post IM/EP). 

We have also added the following note to Table 2: 

Pain assessments at each of the three visits are counted by subject. However, in some cases the same subject reported pain at more than one time or visit. 

7. Lines 230-231: Please clarify if there were 5 participants who experienced "too much" pain after post-electroporation or there were fewer participants with repetitive events. 

- Lines 247-249: Your statement here is not aligned with you statement in lines 229-230 (on the first you say "in the majority of cases" and on the second you say "In every case"). Please include the remaining reported events of "too much pain" in lines 230-232 to ensure consistency. 

Response: Thank you for pointing out this confusion. We have revised (lines 246-54) 258-266 to read:

A total of 16 responses of “too much” pain were reported by 11 subjects over the course of the study, with the majority of these reports at the time of electrical stimulation (11/16=68.5%). One person reported too much pain associated with electrical stimulation at each of the three visits; two others reported too much pain at two of the three electrical stimulations, and four reported too much pain at the time of electrical stimulation once only (two at Visit 1, one at Visit 2, and one at Visit 3). Three subjects who found the pain at the time of electrical stimulation to be acceptable did report too much pain 10 minutes after the injection, with one of these continuing to report pain 30 minutes after the injection. An additional subject reported too much pain 30 minutes after injection at the third visit only.

Acceptability of electroporation

8. You have only analyzed responses from the final visit, after the series of vaccinations had been completed. In a real life scenario it would be important to consider the perceptions after the first vaccination/EP since these may have an impact in retention. Please consider including this analysis and comparing the responses between the first and last visit. Also note that your study participants were under a very controlled research environment with possible extensive counselling which will differ in a real life scenario. 

Response: We thank the reviewer for pointing out the importance of evaluating the first as well as the last visit responses, and have revised this section of the paper (lines 346-357) 368-379 as follows: 

Participants were asked at each visit if they felt that the electroporation procedure would be acceptable if it provided protection from serious disease. After the first vaccination, 44/45 of the participants responded positively, with one uncertain. After the second vaccination, 43/45 responded positively, with one each responding “no” and “uncertain.” After the third vaccination, 43/45 responded positively, with the remaining two responding “no.” Responses were similar when participants were asked if the electroporation procedure would be acceptable if it enhanced scientific knowledge about how best to administer/treat a disease. After the first vaccine, 44/45 responded positively, with one uncertain. After the last vaccine, 42/45 responded positively, with two responding “no” to both questions and one “uncertain.” The two subjects who responded “no”, as well as the subject who was “uncertain” about the second question only, had both reversed previously affirmative answers for both questions, although neither had reported “too much” pain at any time point or visit. 

Also note that your study participants were under a very controlled research environment with possible extensive counselling which will differ in a real life scenario.

The reviewer is correct that this should be noted; we have added the following text to the limitations section of the Discussion section of our paper (lines 421-24). 460-463

In addition, we cannot be certain as to the generalizability of the study, since study participants participated in a controlled research environment with ample opportunities for questions, comments, and explanations of the process. In a real-world situation, it is unlikely that this degree of support would be available.

Discussion: 

9. Please comment on whether the grading of pain at the first visit may have been influenced by the fact that it was a complete new experience and the fear related to the procedure could have also played an important role. 

Response: We thank the reviewer for this suggestion. We have inserted the following into the Discussion section (ll. 381-84): 419-421

This was likely an emotional response to a new and “unknown” procedure, since the physical response, as rated quantitatively, was more often reported to be “intense” at some time point during visits 2 and 3 than at visit 1.

10. Lines 324-325: "... three electroporation procedures and one needle injection". It is one injection per electroporation. Please revise. 

Response: We apologize for this confusion. We were referring to the number of visits in the entire series (four). We have clarified our meaning with the following revision in the text (ll. 387-91) 424-28

No major differences in pain were reported within individuals across visits, and all participants returned as required and completed the entire vaccination series that consisted of a total of four visits: three visits at which the electroporation procedure was used to administer the vaccine, and one visit (the last) at which needle injection was used rather than electroporation.

---

## [Editor Report · Decision Letter 3]

30 Apr 2020

Acceptability and tolerability of repeated intramuscular electroporation of Multi-antigenic HIV (HIVMAG) DNA vaccine among healthy African participants in a phase 1 randomized controlled trial.

PONE-D-19-22032R3

Dear Dr. MPENDO,

We are pleased to inform you that your manuscript has been judged scientifically suitable for publication and will be formally accepted for publication once it complies with all outstanding technical requirements.

With kind regards,

David Joseph Diemert, M.D.

Academic Editor

PLOS ONE
---

## [Editor Report · Acceptance letter]

6 May 2020

PONE-D-19-22032R3 

Acceptability and tolerability of repeated intramuscular electroporation of Multi-antigenic HIV (HIVMAG) DNA vaccine among healthy African participants in a phase 1 randomized controlled trial. 

Dear Dr. Mpendo:

I am pleased to inform you that your manuscript has been deemed suitable for publication in PLOS ONE. Congratulations! Your manuscript is now with our production department. 

With kind regards,

on behalf of

Dr. David Joseph Diemert 

Academic Editor

PLOS ONE